# POGEMA: A BENCHMARK PLATFORM FOR COOPERATIVE MULTI-AGENT PATHFINDING

**Alexey Skrynnik**[✉1,2,3]**, Anton Andreychuk**[✉1]**, Anatolii Borzilov**[2,3]**,
Alexander Chernyavskiy**[3]**, Konstantin Yakovlev**[2,1]**, and Aleksandr Panov**[1,3]

[1]AIRI, Moscow, Russia; [2]FRC CSC RAS, Moscow, Russia; [3]MIPT, Dolgoprudny, Russia

## ABSTRACT

Multi-agent reinforcement learning (MARL) has recently excelled in solving challenging cooperative and competitive multi-agent problems in various environments, typically involving a small number of agents and full observability. Moreover, a range of crucial robotics-related tasks, such as multi-robot pathfinding, which have traditionally been approached with classical non-learnable methods (e.g., heuristic search), are now being suggested for solution using learning-based or hybrid methods. However, in this domain, it remains difficult, if not impossible, to conduct a fair comparison between classical, learning-based, and hybrid approaches due to the lack of a unified framework that supports both learning and evaluation. To address this, we introduce POGEMA, a comprehensive set of tools that includes a fast environment for learning, a problem instance generator, a collection of predefined problem instances, a visualization toolkit, and a benchmarking tool for automated evaluation. We also introduce and define an evaluation protocol that specifies a range of domain-related metrics, computed based on primary evaluation indicators (such as success rate and path length), enabling a fair multi-fold comparison. The results of this comparison, which involves a variety of state-of-the-art MARL, search-based, and hybrid methods, are presented.

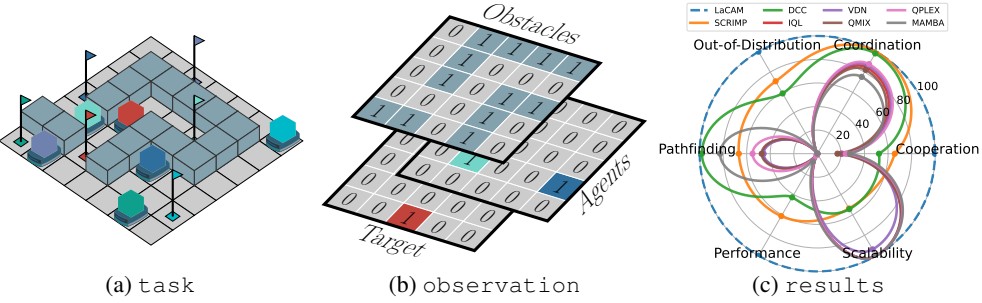

(a) task     (b) observation     (c) results

Figure 1: (a) Example of the multi-agent pathfinding problem considered in POGEMA: each agent must reach its goal, denoted by a flag of the same color. (b) Observation tensor of the red agent. (c) Evaluation results of MARL, hybrid, and search-based solvers on POGEMA benchmark.

## 1 INTRODUCTION

Multi-agent reinforcement learning (MARL) has gained increasing attention recently, with significant progress achieved in the field (Canese et al., 2021; Nguyen et al., 2020; Wong et al., 2023). MARL methods have demonstrated the ability to generate well-performing agents' policies in strategic games (Arulkumaran et al., 2019; Ye et al., 2020), sports simulators (Song et al., 2023; Zang et al., 2024), multi-component robot control (Wang et al., 2024), city traffic control (Kolat et al., 2023), and autonomous driving (Zhou et al., 2020). Several approaches currently exist for formulating

Corresponding authors: ✉skrynnikalexey@gmail.com, ✉andreychuk@airi.net

and solving MARL problems, depending on the information available to agents and the type of communication allowed in the environment (Zhang et al., 2021). With the growing interest in robotic applications, decentralized cooperative learning that minimizes communication between agents has recently attracted particular attention (Singh et al., 2022; Zhang et al., 2020).

A prominent example of an important and practically-inspired problem that can benefit from this type of learning is the so-called multi-agent pathfinding (MAPF) (Stern et al., 2019). In this problem a group of (homogeneous) agents is confined to a graph of locations and at each time step an agent can either wait at the current vertex or move to an adjacent one. Each agent is assigned a goal and the task is to make the agents reach their goals as fast as possible (i.e. using fewer actions) in a safe way, i.e. avoiding the inter-agent collisions as well as collisions with the static obstacles. Numerous practically important applications mimic the discrete nature of MAPF and actively exploit MAPF solutions in real world. A major example is automated warehouses (Dekhne et al., 2019) where robots move synchronously utilizing atomic moves. Furthermore, many works, e.g., (Hönig et al., 2016; Ma et al., 2019; Okumura et al., 2022), describe how MAPF solutions can be transferred to real robots that are subject to kinodynamic constraints, inaccurate execution, and other physically inspired complications. Thus, the MAPF problem serves as a highly useful abstraction that distills the core challenge of any multi-agent navigation problem: coordinating actions between agents to minimize the risk of collisions while optimizing a given cost objective.

Conventional MAPF solvers rely on such algorithmic techniques as systematic (heuristic) search: A*+ID+OD (Standley, 2010), CBS (Sharon et al., 2015), M* (Wagner & Choset, 2011), ICTS (Sharon et al., 2013); or reduce the MAPF problem to the other established computational problem like boolean satisfiability (Surynek et al., 2016) or integer linear programming; or leverage dedicated rule-based techniques (de Wilde et al., 2013). Meanwhile, learning-based MAPF solvers have recently been receiving increased attention, such as (Sartoretti et al., 2019; Ma et al., 2021; Damani et al., 2021; Skrynnik et al., 2024a), to name a few. A key advantage of such solvers is their decentralized nature, allowing each agent to act independently, which can significantly reduce costs compared to classical MAPF solvers requiring centralized control.

From a learning perspective, MAPF is a highly specific problem that differs significantly from well-known MARL problems such as the StarCraft Multi-agent Challenge (SMAC) (Ellis et al., 2024) or Google Research Football (Kurach et al., 2020). In MAPF, generalization to different types of maps and numbers of agents is essential, as real-world MAPF solvers must handle varying map topologies and agent populations. Furthermore, the number of agents in MAPF is often very large – not just dozens (as in SMAC) but hundreds or even thousands of agents are moving simultaneously in the environment. Unsurprisingly, the vast majority of state-of-the-art learnable MAPF solvers, such as (Skrynnik et al., 2023; 2024a;b; Ma et al., 2021; Wang et al., 2023; Sartoretti et al., 2019; Wang et al., 2020a; Liu et al., 2020; Damani et al., 2021), are the hybrid solvers that rely on both traditional search-based techniques and learnable components.

These solvers are developed using different frameworks, environments and datasets and are evaluated accordingly, i.e. in the absence of the unifying evaluation framework, consisting of the evaluation tool, protocol (that defines common performance indicators) and the dataset of the problem instances. Moreover, currently most of the pure MARL methods, i.e. the ones that do not involve search-based modules, such as QMIX (Rashid et al., 2020), MAMBA (Egorov & Shpilman, 2022), MAPPO (Yu et al., 2022) etc., are mostly not included in comparison. This exclusion is likely due to the lack of a unified benchmark that is compatible with or integrated with existing MARL learning frameworks.

To close the mentioned gaps we introduce POGEMA, a comprehensive set of tools that includes:

- a fast and flexible environment supporting different multi-robot pathfinding problems, coupled with a generator of diverse problem instances to facilitate multi-task learning and generalization testing,

- a visualization toolkit enabling the creation of high-quality vector-based plots and animations for enhanced analysis and presentation,

- a benchmarking tool for automated, parallel evaluation of learning-based, planning-based, and hybrid approaches, streamlining comparison across methodologies,

- a standardized evaluation protocol offering domain-specific metrics derived from primary performance indicators, ensuring robust and fair comparisons between methods.

## 2 RELATED WORK

Currently, a huge variety of MARL environments exists that are inspired by various practical applications and encompass a broad spectrum of nuances in problem formulations. Notably, they include a diverse array of computer games (Samvelyan et al., 2019b; Ellis et al., 2024; Rutherford et al., 2023; Carroll et al., 2019; Suarez et al., 2024; Johnson et al., 2016; Bonnet et al., 2023; Baker et al., 2020; Kurach et al., 2020). Additionally, they address complex social dilemmas (Agapiou et al., 2022) including public goods games, resource allocation problems (Papoudakis et al., 2021), and multi-agent coordination challenges. Some are practically inspired, showcasing tasks such as competitive object tracking (Pan et al., 2022), infrastructure management and planning (Leroy et al., 2024), and automated scheduling of trains (Mohanty et al., 2020). Beyond these, the environments simulate intricate, interactive systems such as traffic management and autonomous vehicle coordination (Vinitsky et al., 2022), multi-agent control tasks (Rutherford et al., 2023; Peng et al., 2021), and warehouse management (Gupta et al., 2017). Each scenario is designed to challenge and analyze the collaborative and competitive dynamics that emerge among agents in varied and complex contexts. We summarize the most wide-spread MARL environments in Table 1. A detailed description of each column is presented below.

| Environment | Repository | Navigation | Partially observable | Python based | Hardware-agnostic setup | Performance >10K Steps/s | Procedural generation | Requires generalization | Evaluation protocols | Tests & CI | PyPi Listed | Scalability >1000 Agents | Induced behavior |
|---|---|---|---|---|---|---|---|---|---|---|---|---|---|
| Flatland (Mohanty et al., 2020) | link | ✓ | ✓ | ✓ | ✓ | ✗ | ✓ | ✓ | ✓ | ✗ | ✓ | ✓ | Coop |
| Gigastep (Lechner et al., 2024) | link | ✓ | ✓ | ✗ | ✗ | ✓ | ✗ | ✗ | ✗ | ✗ | ✓ | ✓ | Mixed |
| GoBigger (Zhang et al., 2023) | link | ✓ | ✓ | ✓ | ✓ | ✗ | ✗ | ✗ | ✓ | ✗ | ✓ | ✗ | Mixed/Coop |
| Google Research Football (Kurach et al., 2020) | link | ✓ | ✓ | ✗ | ✗ | ✗ | ✗ | ✗ | ✗ | ✓ | ✗ | ✗ | Mixed |
| Griddly (Bamford, 2021) | link | ✓ | ✓ | ✗ | ✗ | ✓ | ✓ | ✗ | ✗ | ✓ | ✓ | ✓ | Mixed |
| Hide-and-Seek (Baker et al., 2020) | link | ✓ | ✓ | ✓ | ✗ | ✗ | ✗ | ✗ | ✗ | ✗ | ✗ | ✗ | Comp |
| IMP-MARL (Leroy et al., 2024) | link | ✗ | ✓ | ✓ | ✓ | ✗ | ✗ | ✗ | ✓ | ✗ | ✗ | ✓ | Coop |
| Jumanji (XLA) (Bonnet et al., 2023) | link | ✓ | ✓ | ✓ | ✗ | ✓ | ✗ | ✗ | ✓ | ✓ | ✓ | ✗ | Mixed |
| LBF (Papoudakis et al., 2021) | link | ✓ | ✓ | ✓ | ✓ | ✗ | ✗ | ✗ | ✗ | ✓ | ✓ | ✗ | Coop |
| MAMuJoCo (Peng et al., 2021) | link | ✗ | ✓ | ✓ | ✓ | ✗ | ✗ | ✗ | ✗ | ✓ | ✓ | ✗ | Coop |
| Magent (Zheng et al., 2018) | link | ✓ | ✓ | ✗ | ✗ | ✓ | ✗ | ✗ | ✗ | ✓ | ✗ | ✓ | Mixed |
| MATE (Pan et al., 2022) | link | ✓ | ✓ | ✓ | ✓ | ✗ | ✗ | ✗ | ✓ | ✓ | ✗ | ✗ | Coop |
| MeltingPot (Agapiou et al., 2022) | link | ✓ | ✓ | ✗ | ✗ | ✗ | ✗ | ✓ | ✓ | ✓ | ✓ | ✗ | Mixed/Coop |
| MALMO (Johnson et al., 2016) | link | ✓ | ✓ | ✗ | ✗ | ✗ | ✓ | ✓ | ✓ | ✓ | ✗ | ✗ | Mixed |
| MPE (Lowe et al., 2017) | link | ✓ | ✓ | ✓ | ✓ | ✓ | ✗ | ✗ | ✗ | ✗ | ✗ | ✗ | Mixed |
| MPE (XLA) (Rutherford et al., 2023) | link | ✓ | ✓ | ✓ | ✗ | ✓ | ✗ | ✗ | ✗ | ✓ | ✓ | ✗ | Mixed |
| Multi-agent Brax (XLA) (Rutherford et al., 2023) | link | ✗ | ✓ | ✓ | ✗ | ✓ | ✗ | ✗ | ✗ | ✓ | ✓ | ✗ | Coop |
| Multi-Car Racing (Schwarting et al., 2021) | link | ✓ | ✓ | ✓ | ✓ | ✗ | ✗ | ✗ | ✗ | ✗ | ✗ | ✗ | Comp |
| Neural MMO (Suarez et al., 2024) | link | ✓ | ✓ | ✓ | ✗ | ✗ | ✓ | ✗ | ✓ | ✓ | ✓ | ✓ | Comp |
| Nocturne (Vinitsky et al., 2022) | link | ✓ | ✓ | ✗ | ✗ | ✗ | ✗ | ✗ | ✓ | ✓ | ✗ | ✓ | Mixed |
| Overcooked (Carroll et al., 2019) | link | ✓ | ✗ | ✓ | ✓ | ✗ | ✗ | ✓ | ✓ | ✓ | ✓ | ✗ | Coop |
| Overcooked (XLA) (Rutherford et al., 2023) | link | ✓ | ✗ | ✓ | ✗ | ✓ | ✗ | ✓ | ✗ | ✓ | ✓ | ✓ | Coop |
| RWARE (Papoudakis et al., 2021) | link | ✓ | ✓ | ✓ | ✓ | ✓ | ✗ | ✗ | ✗ | ✓ | ✓ | ✗ | Coop |
| SISL (Gupta et al., 2017) | link | ✓ | ✓ | ✓ | ✓ | ✓ | ✗ | ✗ | ✗ | ✓ | ✓ | ✗ | Coop |
| SMAC (Samvelyan et al., 2019b) | link | ✓ | ✓ | ✗ | ✗ | ✗ | ✗ | ✗ | ✓ | ✗ | ✗ | ✗ | Mixed/Coop |
| SMAC v2 (Ellis et al., 2024) | link | ✓ | ✓ | ✗ | ✗ | ✗ | ✗ | ✗ | ✓ | ✗ | ✗ | ✗ | Mixed/Coop |
| SMAX (XLA) (Rutherford et al., 2023) | link | ✓ | ✓ | ✓ | ✗ | ✓ | ✗ | ✗ | ✗ | ✓ | ✓ | ✓ | Mixed/Coop |
| POGEMA (ours) | link | ✓ | ✓ | ✓ | ✓ | ✓ | ✓ | ✓ | ✓ | ✓ | ✓ | ✓ | Mixed |

Table 1: Comparison of different multi-agent reinforcement learning environments.

**Navigation** Navigation tasks arise in almost all multi-agent environments (e.g. unit navigation in SMAC or robotic warehouse management in RWARE), however only a handful of environments specifically focus on challenging navigation problems: Flatland, Nocturne, RWARE, and POGEMA.

**Partially observable**    Partial observability is an intrinsic feature of a generic multi-agent problem, meaning that an individual agent does not have access to the full state of the environment but rather is able to observe it only locally (e.g. an agent is able to determine the locations of the other agents and/or static obstacles only in its vicinity). Most of the considered environments are partially observable, with the exception of Overcooked.

**Environment Setup    Python based** means the environment is implemented in Python, unlike many other multi-agent environments that include bindings for other languages or external dependencies, which can complicate usage. Pure Python implementations ensure ease of modification and customization, allowing researchers to easily adapt and extend the environments. **Hardware-agnostic setup** means the environment doesn't require any specific type of hardware for training or inference, offering flexibility across different systems.

**Performance >10K Steps/s**    Training and evaluating multi-agent reinforcement learning agents often requires making billions of steps (transitions) in the environment. Thus, it is crucial that each transition is computed efficiently. In general, performing more than 10K steps per second is a good indicator of the environment's efficiency. While XLA versions can provide high performance by vectorizing the environment on GPU, they require modern hardware setups, which can be a barrier for some researchers. In contrast, fast environments like POGEMA or RWARE can achieve high performance without such stringent hardware requirements, making them more accessible and easier to integrate into a variety of research projects.

**Procedural generation**    To improve the ability of RL agents to solve problem instances that are not the same that were used for training (the so-called ability to generalize) procedural generation of the problem instances is commonly used. I.e. the environment does not rely on a predefined set of training instances but rather procedurally generates them to prevent overfitting. As highlighted in the Procgen paper (Cobbe et al., 2020), this approach ensures that agents develop robust strategies capable of adapting to novel and diverse situations. Moreover, in multi-agent settings, agents must be able to handle and adapt to a variety of unforeseen agent behaviors and strategies, ensuring robustness and flexibility in dynamic environments (Agapiou et al., 2022).

**Evaluation protocols**    means that the environment features a comprehensive evaluation API, including computation of distinct performance indicators and visualization tools. These capabilities allow precise performance measurement and deeper insights into RL agents' behavior, going beyond just reward curves, which can often hide agents exploiting the reward system rather than genuinely solving the tasks.

**Tests and CI** means the environment is set up for development with continuous integration and is covered with tests, which are essential for collaborative open-source development. **PyPI listed** indicates that the environment library is listed on PyPI[1], making it easy to install and integrate into projects with a simple `pip install` command. **Scalability to >1000 Agents** refers to the environment's ability to handle over 1000 simultaneously acting agents, ensuring robust performance and flexibility for large-scale multi-agent simulations.

**Induced behavior**    Multi-agent behavior can be influenced by modifying the reward function (Shoham & Leyton-Brown, 2008). Competitive (Comp) behavior arises when a joint strategy benefits one player but disadvantages others. In a two-player game, this corresponds to a Pareto-efficient outcome. Minimax games, where agents' rewards sum to zero, are classic examples of competitive games. Cooperative (Coop) behavior (Du et al., 2023; Shoham & Leyton-Brown, 2008) occurs when agents share a unified reward function or pursue the same goal, rewarded only by its completion. Social dilemmas are a key example of cooperation. Mixed behavior (Littman, 1994) doesn't limit the agents' objectives or interactions, blending cooperative and competitive behaviors. A well-known example is the iterated prisoner's dilemma.

As we aim to create a lightweight and easy-to-configure multi-agent environment for reinforcement learning and pathfinding tasks, we consider the following factors essential. First and foremost, our environment is fully compatible with the native Python API: we target pure Python builds independent of

---

[1]https://pypi.org

hardware-specific software with a minimal number of external dependencies. Moreover, we underline the importance of constant extension and flexibility of the environment. Thus, we prioritize testing and continuous integration as cornerstones of the environment, as well as trouble-free modification of the transition dynamics. Secondly, we highlight that our environment targets generalization and may utilize procedural generation. Last but not least, we target high computational throughput (i.e., the number of environment steps per second) and robustness to an extremely large number of agents (i.e., the environment remains efficient under high loads). The detailed overview of the mentioned related papers is provided in Appendix F.

Despite the diversity of available environments, most research papers tend to utilize only a selected few. Among these, the most popular are the StarCraft Multi-agent Challenge (SMAC), Multi-agent MuJoCo (MAMuJoCo), and Google Research Football (GRF), with SMAC being the most prevalent in top conference papers. The popularity of these environments is likely due to their effective contextualization of algorithms. For instance, to demonstrate the advantages of a method, it is crucial to test it within a well-known environment.

The evaluation protocols in these environments typically feature learning curves that highlight the performance of each algorithm under specific scenarios. For SMAC, these scenarios involve games against predefined bots with specific units on both sides. In MAMuJoCo, the standard tasks involve agents controlling different sets of joints, while in GRF, the scenarios are games against predefined policies from Football Academy scenarios. Proper evaluation of MARL approaches is a serious concern. For SMAC, it's highlighted in the paper (Gorsane et al., 2022), which proposes a unified evaluation protocol for this benchmark. This protocol includes default evaluation parameters, performance metrics, uncertainty quantification, and a results reporting scheme.

The variability of results across different studies underscores the importance of a well-defined evaluation protocol, which should be developed alongside the presentation of the environment. In our study, we provide not only the environemnt but also the evaluation protocol, popular MARL baselines, and modern learnable MAPF approaches to better position our benchmark within the context.

## 3 POGEMA

POGEMA, which comes from Partially-Observable Grid Environment for Multiple Agents, is an umbrella name for a collection of versatile and flexible tools aimed at developing, debugging and evaluating different methods and policies tailored to solve several types of multi-agent pathfinding tasks. The source code is available at: POGEMA Benchmark[2], POGEMA Toolbox[3] and POGEMA Environment[4].

### 3.1 POGEMA ENVIRONMENT

POGEMA [5] environment is a core of POGEMA suite. It implements the basic mechanics of agents' interaction with the world. The environemnt is open-sourced under MIT license. POGEMA provides integration with existing RL frameworks: PettingZoo (Terry et al., 2021), PyMARL (Samvelyan et al., 2019a), and Gymnasium (Towers et al., 2023).

**Basic mechanics**  The workspace where the agents navigate is represented as a grid composed of blocked and free cells. Only the free cells are available for navigation. At each timestep each agent individually and independently (in accordance with a policy) picks an action and then these actions are performed simultaneously. POGEMA implements collision shielding mechanism, i.e. if an agent picks an action that leads to an obstacle (or out-of-the-map) than it stays put, the same applies for two or more agents that wish to occupy the same cell. POGEMA also has an option when one of the agents deciding to move to the common cell does it, while the others stay where they were. The episode ends when the predefined timestep, episode length, is reached. The episode can also end before this timestep if certain conditions are met, i.e. all agents reach their goal locations if MAPF problem (see below) is considered.

---

[2]https://github.com/Cognitive-AI-Systems/pogema-benchmark
[3]https://github.com/Cognitive-AI-Systems/pogema-toolbox
[4]https://github.com/Cognitive-AI-Systems/pogema
[5]https://pypi.org/project/pogema

**Problem settings**   POGEMA supports two generic types of multi-agent navigation problems. In the first variant, dubbed MAPF (from Multi-agent Pathfinding), each agent is provided with the unique goal location and has to reach it avoiding collisions with the other agents and static obstacles. For MAPF problem setting POGEMA supports both *stay-at-target* behavior (when the episode successfully ends only if all the agents are at their targets) and *disappear-at-target* (when the agent is removed from the environment after it first reaches its goal). The second variant is a *lifelong* version of multi-agent pathfinding and is dubbed accordingly – LMAPF. Here each agent upon reaching a goal is immediately assigned another one (not known to the agent beforehand). Thus, the agents are constantly moving through in the environment until episode ends.

**Observation**   At each timestep each agent in POGEMA receives an individual ego-centric observation represented as a tensor – see Figure 1. The latter is composed of the following $(2R+1)\times(2R+1)$ binary matrices, where $R$ is the observation radius set by the user:

1. Static Obstacles – 0 means the free cell, 1 – static obstacle

2. Other Agents – 0 means no agent in the cell, 1 – the other agent occupies the cell

3. Targets – projection of the (current) goal location of the agent to the boundary of its field-of-view

The suggested observation, which is, indeed, minimalist and simplistic, can be modified by the user using wrapper mechanisms. For example, it is not uncommon in the MAPF literature to augment the observation with additional matrices encoding the agent's path-to-goal (constructed by some global pathfinding routine) (Skrynnik et al., 2024a) or other variants of global guidance (Ma et al., 2021).

**Reward**   POGEMA features the most intuitive and basic reward structure for learning. I.e. an agent is rewarded with $+1$ if it reaches the goal and receives $0$ otherwise. For MARL policies that leverage centralized training a shared reward is supported, i.e. $r_t = goals/agents$ where $goals$ is the number of goals reached by the agents at timestep $t$ and $agents$ is the number of agents. Indeed, the user can specify its own reward using wrappers.

**Performance indicators**   The following performance indicators are considered basic and are tracked in each episode. For MAPF they are: *Sum-of-costs* (*SoC*) and *makespan*. The former is the sum of time steps (across all agents) consumed by the agents to reach their respective goals, the latter is the maximum over those times. The lower those indicators are the more effectively the agents are solving MAPF tasks. For LMAPF the primary tracked indicator is the *throughput* which is the ratio of the number of the accomplished goals (by all agents) to the episode length. The higher – the better.

## 3.2   POGEMA Toolbox

The POGEMA Toolbox is a comprehensive framework designed to facilitate the testing of learning-based approaches within the POGEMA environment. This toolbox offers a unified interface that enables the seamless execution of any learnable MAPF algorithm in POGEMA. Firstly, the toolbox provides robust management tools for custom maps, allowing users to register and utilize these maps effectively within POGEMA. Secondly, it enables the concurrent execution of multiple testing instances across various algorithms in a distributed manner, leveraging Dask[6] for scalable processing. The results from these instances are then aggregated for analysis. Lastly, the toolbox includes visualization capabilities, offering a convenient method to graphically represent the aggregated results through detailed plots. This functionality enhances the interpretability of outcomes, facilitating a deeper understanding of algorithm performance.

POGEMA Toolbox offers a dedicated tool for map generation, allowing the creation of three distinct types of maps: random, mazes and warehouse maps. All generators facilitates map creation using adjustable parameters such as width, height, and obstacle density. The maze generator was implemented based on the generator provided in  (Damani et al., 2021).

---

[6]https://github.com/dask/dask

## 3.3 BASELINES

POGEMA integrates a variety of MARL, hybrid and planning-based algorithms with the environment. These algorithms, recently presented, demonstrate state-of-the-art performance in their respective fields. Table 2 highlights the differences between these approaches. Some, such as LaCAM and RHCR, are centralized search-based planners. Other approaches, such as SCRIMP and DCC, while decentralized, still require communication between agents to resolve potential collisions. Learnable modern approaches for LifeLong MAPF that do not utilize communication include Follower (Skrynnik et al., 2024a), MATS-LP (Skrynnik et al., 2024b), and Switchers (Skrynnik et al., 2023). All these approaches utilize independent PPO (Schulman et al., 2017) as the training method.

| Algorithm | Decentralized | Partial Observability | Fully Integrated into POGEMA | Supports MAPF | Supports LifeLong MAPF | No Global Obstacles Map | No Communication | Parameter Sharing | Decentralized Learning | Model-Based | No Imitation Learning |
|---|---|---|---|---|---|---|---|---|---|---|---|
| MAMBA (Egorov & Shpilman, 2022) | ✓ | ✓ | ✓ | ✓ | ✓ | ✗ | ✗ | ✓ | ✗ | ✓ | ✓ |
| QPLEX (Wang et al., 2020b) | ✓ | ✓ | ✓ | ✓ | ✓ | ✗ | ✓ | ✓ | ✗ | ✗ | ✓ |
| IQL (Tan, 1993) | ✓ | ✓ | ✓ | ✓ | ✓ | ✗ | ✓ | ✓ | ✓ | ✗ | ✓ |
| VDN (Sunehag et al., 2018) | ✓ | ✓ | ✓ | ✓ | ✓ | ✗ | ✓ | ✓ | ✗ | ✗ | ✓ |
| QMIX (Rashid et al., 2020) | ✓ | ✓ | ✓ | ✓ | ✓ | ✗ | ✓ | ✓ | ✗ | ✗ | ✓ |
| Follower (Skrynnik et al., 2024a) | ✓ | ✓ | ✓ | ✗ | ✓ | ✗ | ✓ | ✓ | ✓ | ✗ | ✓ |
| MATS-LP (Skrynnik et al., 2024b) | ✓ | ✓ | ✓ | ✗ | ✓ | ✗ | ✓ | ✓ | ✓ | ✓ | ✓ |
| Switcher (Skrynnik et al., 2023) | ✓ | ✓ | ✓ | ✗ | ✓ | ✓ | ✓ | ✓ | ✓ | ✗ | ✓ |
| SCRIMP (Wang et al., 2023) | ✓ | ✓ | ✗ | ✓ | ✗ | ✗ | ✗ | ✓ | ✗ | ✗ | ✗ |
| DCC (Ma et al., 2021) | ✓ | ✓ | ✓ | ✓ | ✗ | ✗ | ✗ | ✓ | ✗ | ✗ | ✗ |
| LaCAM (Okumura, 2024; 2023) | ✗ | ✗ | ✗ | ✓ | ✗ | ✗ | - | - | - | - | - |
| RHCR (Li et al., 2021b) | ✗ | ✗ | ✗ | ✗ | ✓ | ✗ | - | - | - | - | - |

Table 2: This table provides an overview of various baseline approaches supported by POGEMA and their features in the context of decentralized multi-agent pathfinding.

The following modern MARL algorithms are included as baselines: MAMBA (Egorov & Shpilman, 2022), QPLEX (Wang et al., 2020b), IQL (Tan, 1993), VDN (Sunehag et al., 2018), and QMIX (Rashid et al., 2020). For environment preprocessing, we used the scheme provided in the Follower approach, enhancing it with the anonymous targets of other agents' local observations for MAPF scenarios. We utilized the official implementation of MAMBA, as provided by its authors[7], and employed PyMARL2 framework[8] for establishing MARL baselines. We used the default parameters for MAMBA, since Dreamer (which serves as the foundation for MAMBA) is known to work effectively across domains with nearly the same hyperparameters. For the other MARL approaches, we performed a hyperparameter sweep over the learning rate, batch size, replay buffer size, and GRU hidden state size, using the best parameters based on the performance scores from the training scenarios on the `Random` and `Mazes` maps.

## 4 EVALUATION PROTOCOL

### 4.1 DATASET

We include the maps of the following types in our evaluation dataset (with the intuition that different maps topologies are necessary for proper assessment):

- `Mazes` – maps that encounter prolonged corridors with 1-cell width that require high level of cooperation between the agents. These maps are procedurally generated.

- `Random` – one of the most commonly used type of maps, as they are easy to generate and allow to avoid overfitting to some special structure of the map. POGEMA contains an integrated random maps generator, that allows to control the density of the obstacles.

---

[7] https://github.com/jbr-ai-labs/mamba
[8] https://github.com/hijkzzz/pymarl2

- `Warehouse` – this type of maps are usually used in the papers related to LifeLong MAPF. While there is no narrow passages, high density of the agents might significantly reduce the overall throughput, especially when agents are badly distributed along the map.

- `Cities` – a set of city maps from MovingAI – the existing benchmark widely used in heuristic-search community Sturtevant (2012). The contained maps have a varying structure and $256 \times 256$ size. It can be used to show how the approach deals with single-agent pathfinding and also deals with the maps that have out-of-distribution structure.

- `Cities-tiles` – a modified `Cities` set of maps. Due to the large size of the original maps, it's hard to get high density of the agents on them. To get more crowded maps, we slice the original maps on 16 pieces with $64 \times 64$ size.

- `Puzzles` – a set of small hand-crafted maps that contains some difficult patterns that mandate the cooperation between that agents.

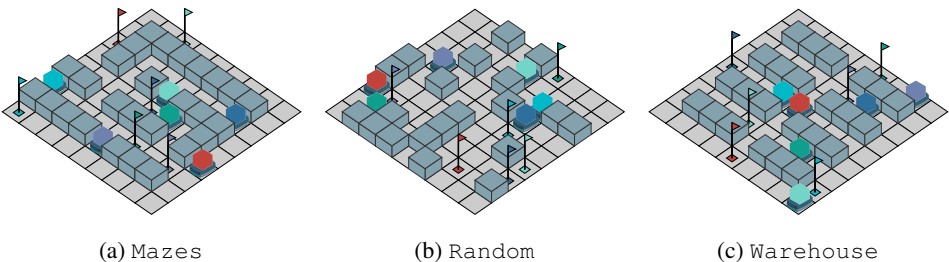

| (a) `Mazes` | (b) `Random` | (c) `Warehouse` |

Figure 2: Examples of map generators presented in POGEMA.

Start and goal locations are generated via random generators. They are generated with fixed seeds, thus can be reproduced. It is guaranteed that each agent has its own goal location and the path to it from its start location exists. Examples of the maps are presented in Figure 2.

## 4.2 METRICS

The existing works related to solving MAPF problems evaluates the performance by two major criteria – success rate and the primary performance indicators mentioned above: sum-of-costs, makespan, throughput. These are directly obtainable from POGEMA. While these metrics allow to evaluate the algorithms at some particular instance, it might be difficult to get a high-level conclusion about the performance. Thus, we want to introduce high-level metrics that cover multiple different aspects:

`Performance` – how well the algorithm works compared to other approaches. To compute this metric we run the approaches on a set of maps similar to the ones, used during training, and compare the obtained results with the best ones.

$$Performance_{MAPF} = \begin{cases} SoC_{best}/SoC \\ 0 \text{ if not solved} \end{cases} \tag{1}$$

$$Performance_{LMAPF} = throughput/throughput_{best} \tag{2}$$

`Out-of-Distribution` – how well the algorithm works on out-of-distribution maps. This metric is computed in the same way as `Performance`, with the only difference that the approaches are evaluated on a set of maps, that were not used during training phase and have different structure of obstacles. For this purpose we utilize maps from `Cities-tiles` set of maps.

$$Out\_of\_Distribution_{MAPF} = \begin{cases} SoC_{best}/SoC \\ 0 \text{ if not solved} \end{cases} \tag{3}$$

$$Out\_of\_Distributionn_{LMAPF} = throughput/throughput_{best} \tag{4}$$

`Cooperation` – how well the algorithm is able to resolve complex situations. To evaluate this metric we run the algorithm on `Puzzles` set of maps.

$$Cooperation_{MAPF} = \begin{cases} SoC_{best}/SoC \\ 0 \text{ if not solved} \end{cases} \tag{5}$$

$$Cooperation_{LMAPF} = throughput/throughput_{best} \tag{6}$$

Scalability – how well the algorithm scales to large number of agents. To evaluate how well the algorithm scales to large number of agents, we run it on a large warehouse map with increasing number of agents and compute the ratio between runtimes with various number of agents.

$$Scalability = \frac{runtime(agents_1)/runtime(agents_2)}{|agents_1|/|agents_2|} \tag{7}$$

Coordination – how well the algorithm avoids collisions between agents and obstacles. Majority of learning-based approaches are decentralized, so there may be cases where multiple agents attempt to occupy the same cell or traverse the same edge simultaneously. There also might be the case when an agent tries to move to the blocked cell. The fewer collisions that occur during the episode, the better. To compute this metric, we use the results obtained on the Mazes and Random sets of maps.

$$Coordination = 1 - \frac{occured\_collisions}{|agents| \times episode\_length} \tag{8}$$

Pathfinding – how well the algorithm works in case of presence of a single agent on a large map. This metric is tailored to determine the ability of the approach to effectively lead agents to their goal locations. For this purpose we run the approaches on large maps from Cities benchmark sets. The closer the costs of the found paths to the optimal ones – the higher the score.

$$Pathfinding = \begin{cases} path\_cost/path\_cost_{optimal} \\ 0 \text{ if path not found} \end{cases} \tag{9}$$

First three metrics, i.e. Performance, Out-of-Distribution, and Cooperation, have the same formula but differ in the set of maps used to compute them, while the remaining three metrics, i.e. Scalability, Coordination, and Pathfinding, are tailored to specific aspects.

### 4.3 EXPERIMENTAL RESULTS

We have evaluated all the supported baselines (12 in total) on both MAPF and LMAPF setups on all 6 datasets. The results of this evaluation are presented in Figure 3. The details about number of maps, number of agents, seeds, etc. are given in the supplementary material (as well as details on how our results can be reproduced). In both setups, i.e., MAPF and LMAPF, the best results in terms of cooperation, out-of-distribution, and performance metrics were obtained by centralized planners, i.e. LaCAM and RHCR respectively.

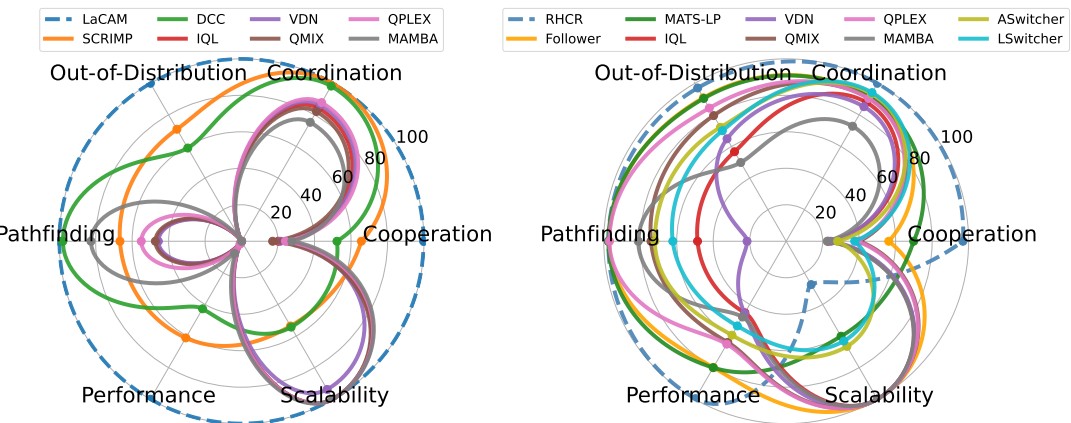

Figure 3: Evaluation of baselines available in POGEMA on (a) MAPF (b) LMAPF instances.

For MAPF tasks, LaCAM notably outperforms all other approaches. Specialized learnable MAPF approaches, such as DCC and SCRIMP, perform better than any of the MARL approaches. When comparing these two specialized approaches, SCRIMP outperforms DCC across such metrics as Performance, Cooperation, and Out-of-Distribution. Surprisingly, SCRIMP underperforms on pathfinding tasks, revealing a weakness in single-agent scenarios that don't require communication, which may represent an out-of-distribution case for this algorithm. It is also worth noting that SCRIMP incorporates an integrated tie-breaking mechanism that ensures collision-free actions. MARL algorithms such as QPLEX, VDN, and QMIX perform significantly worse than other approaches, showing a substantial gap in the results. This can be attributed to the absence of additional techniques used in hybrid approaches. This may suggest that the MARL community lacks large-scale approaches and benchmarks for these tasks. Among MARL approaches, MAMBA achieves the best results in performance, cooperation, and pathfinding metrics, which can be attributed to its communication mechanism. However, its performance remains much worse than that of specialized methods, and it fails to solve any instances in the out-of-distribution dataset.

For LMAPF tasks, the centralized approach, RHCR, is superior in all cases except for the scalability metric. Among the non-centralized approaches, the best results, depending on the metric, are shown by either MATS-LP or Follower. The only metric in which these approaches differ significantly is the cooperation metric, where MATS-LP performs better than Follower. However, MATS-LP requires considerably more test-time computation, as it runs MCTS for each agent at every step (see Appendix I for more details). Additionally, there are two hybrid approaches – ASwitcher and LSwitcher – that differ in how they alternate between planning-based and learning-based components. One reason for their mediocre performance is the lack of global information, i.e., they assume that agents have no prior knowledge of the map, requiring each agent to reconstruct it based on the local observations. Differently from the MAPF scenarios, MARL approaches can compete with hybrid methods on LMAPF instances. This behavior is partly explained by the use of Follower's observation model, which is specifically designed to solve LMAPF. Among MARL approaches, QPLEX delivers the best results, in contrast to MAPF tasks it even outperforming MAMBA.

## 5 Conclusion, Limitations and Future Work

This paper presents POGEMA – a powerful suite of tools tailored for creating, assessing, and comparing methods and policies for multi-agent pathfinding. POGEMA encompasses a fast learning environment and a comprehensive evaluation toolbox suitable for pure MARL, hybrid, and search-based solvers. It includes a wide array of methods as baselines. The evaluation protocol described, along with a rich set of metrics, assists in assessing the generalization and scalability of all approaches. Visualization tools enable qualitative examination of algorithm performance. Integration with the well-known MARL API and map sets facilitates the benchmark's expansion. Existing limitations are two-fold. First, a conceptual limitation is that communication between agents is not currently disentangled in POGEMA environment. Second, technical limitations include the lack of JAX support and integration with the other GPU parallelization tools.

Future work could explore large-scale training setups for MARL methods, capable of handling large agent populations and scenarios, particularly within the CTDE (Centralized Training, Decentralized Execution) paradigm. Second, advancing communication learning suited to large-scale settings, where local interactions are crucial, is another promising direction. Third, addressing memory limitations by developing efficient approaches for long-horizon tasks without relying on global guidance is also vital. Furthermore, leveraging POGEMA's procedural map generation and expert data from centralized solvers could enhance imitation learning and aid in training decentralized foundation models for MAPF. Finally, studying heterogeneous policy coordination may enable effective collaboration among diverse agents and advance concurrent training of different algorithms in shared environments.

## 6 Acknowledgments

We thank the researchers using POGEMA in their studies, as well as the repository users who contributed issues, questions, and feedback to improve the platform. Our thanks also go to the anonymous reviewers for their valuable comments and suggestions, which enhanced the quality of this work.

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

## Appendix Contents

## A  Evaluation Setup Details

POGEMA benchmark contains 6 different sets of maps and all baseline approaches were evaluated on them either on MAPF or on LMAPF instances. Regardless the type of instances, number of maps, seeds and agents were the same. Table 3 contains all information about these numbers. In total, this corresponds to 3,376 episodes for each scenario type. Note that there is no MaxSteps (LMAPF) value for Cities set of maps. This set of maps was used only for pathfinding meta-metric, thus all approaches were evaluated only on MAPF instances with a single agent.

We used implementation LaCAM-v3[9], RHCR[10]. For learning-based approaches beyond MARL, we used their official implementations and the provided weights for Follower[11], MATS-LP[12], Switcher[13], SCRIMP[14], and DCC[15].

| | Agents | Maps | MapSize | Seeds | MaxSteps (MAPF) | MaxSteps (LMAPF) |
|---|---|---|---|---|---|---|
| Random | [8, 16, 24, 32, 48, 64] | 128 | 17×17 - 21×21 | 1 | 128 | 256 |
| Mazes | [8, 16, 24, 32, 48, 64] | 128 | 17×17 - 21×21 | 1 | 128 | 256 |
| Warehouse | [32, 64, 96, 128, 160, 192] | 1 | 33×46 | 128 | 128 | 256 |
| Puzzles | [2, 3, 4] | 16 | 5×5 | 10 | 128 | 256 |
| Cities | [1] | 8 | 256×256 | 10 | 2048 | - |
| Cities-tiles | [64, 128, 192, 256] | 128 | 64×64 | 1 | 256 | 256 |

Table 3: Details about the instances on different sets of maps.

## B  Results for MAPF Benchmark

In this section, we present the extended results of the MAPF benchmark analysis, highlighting the performance, out-of-distribution handling, scalability, cooperation, coordination, and pathfinding

---

[9]https://github.com/Kei18/lacam3
[10]https://github.com/Jiaoyang-Li/RHCR
[11]https://github.com/AIRI-Institute/learn-to-follow
[12]https://github.com/CognitiveAISystems/mats-lp
[13]https://github.com/AIRI-Institute/when-to-switch
[14]https://github.com/marmotlab/SCRIMP
[15]https://github.com/ZiyuanMa/DCC

capabilities of various approaches. The experiments were conducted on different map types and sizes, employing metrics such as SoC, CSR, and makespan to evaluate effectiveness. Detailed visual and tabular data illustrate how centralized and learnable approaches perform under various conditions.

> Please note that the previous arXiv version contains different SCRIMP results due to a technical error, which we discovered while refactoring the code for the camera-ready version.

## B.1 PERFORMANCE

The performance metrics were calculated using `Mazes` and `Random` maps of size close to $20 \times 20$. The primary metrics here are SoC and CSR. The results of all the MAPF approaches over different numbers of agents are presented in Figure 4. The superior performance is shown by the centralized approach, LaCAM. Next best-performing approach is SCRIMP that is substantially outperforms another specialized solver – DCC. Among the MARL methods, better results are shown by MAMBA for both metrics. However, it narrowly lags behind the specialized approaches, DCC and SCRIMP.

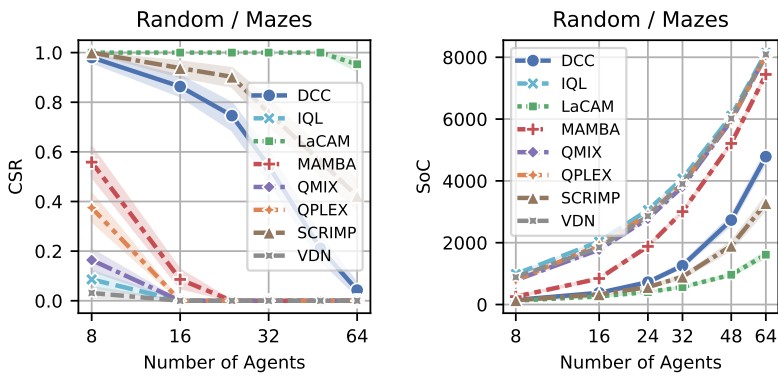

Figure 4: Performance of MAPF approaches on `Random` and `Mazes` maps, based on CSR (higher is better) and SoC (lower is better) metrics. The shaded area indicates $95\%$ confidence intervals.

## B.2 OUT-OF-DISTRIBUTION

Out-of-Distribution metric was calculated on `Cities-tiles` dataset, that consists of pieces of cities maps with $64 \times 64$ size. Due to much larger size compared to `Mazes` and `Random` maps, the amount of agents was also significantly increased. The results are presented in Figure 5. Here again centralized search-based planner, i.e. LaCAM, demonstrates the best results both in terms of CSR and SoC. Among hybrid methods, SCRIMP demonstrates better performance in terms of CSR, however, average SoC of its solutions is almost the same as solutions found by DCC. MARL approaches are unable to solve any instance even with 64 agents.

## B.3 SCALABILITY

The results of how well the algorithm scales with a large number of agents are shown in Figure 6. The experiments were conducted on a `warehouse` map. The plot is log-scaled. The best scalability is achieved with the centralized LaCAM approach, which is a high-performance approach. DCC demonstrates the worst results in terms of runtime; however, regarding the Scalability metric, DCC's results are identical to SCRIMP's. Despite an initially high runtime, the scalability of MAMBA is better than other approaches; however, this could be attributed to the high cost of GPU computation, which is due to the large number of parameters in the neural network and is the limiting factor of this approach.

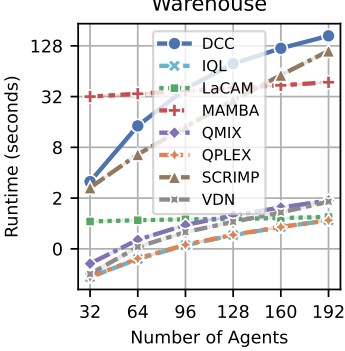

Figure 6: Runtime in seconds for each algorithm. The plot is log-scaled.

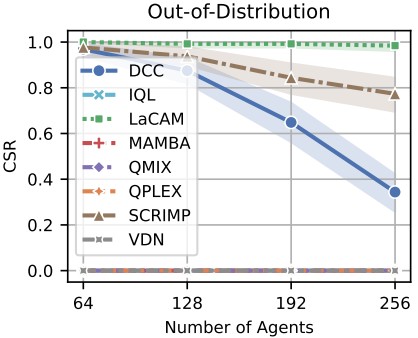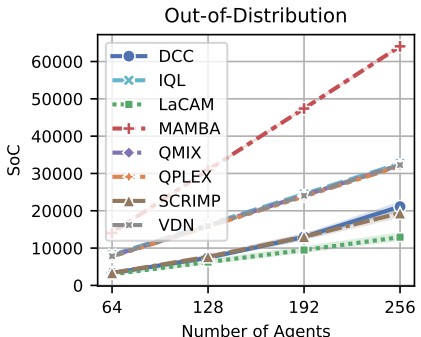

Figure 5: Performance of MAPF approaches on `Cities-tiles` maps. These results were utilized to compute Out-of-Distribution metric. The shaded area indicates 95% confidence intervals.

### B.4 COOPERATION

How well the algorithm is able to resolve complex situations on the `Puzzles` set is reflected in the results presented in Table 4. The only approach out of the evaluated ones that is able to solve all the instances in this set is LaCAM. Out of the rest approaches, as well as on other sets of instances, the best results is demonstrated by SCRIMP, which outperforms DCC both in terms of CSR and SoC. Among MARL approaches, better cooperation is demonstrated by QMIX, outperforming QPLEX, VDN, IQL, and even MAMBA.

| Algorithm | CSR | SoC |
|---|---|---|
| DCC | $0.74 \pm 0.04$ | $93.50 \pm 10.47$ |
| IQL | $0.47 \pm 0.04$ | $250.03 \pm 19.09$ |
| LaCAM | $1.00$ | $20.84 \pm 1.65$ |
| MAMBA | $0.40 \pm 0.05$ | $173.60 \pm 14.04$ |
| QMIX | $0.35 \pm 0.04$ | $253.28 \pm 16.04$ |
| QPLEX | $0.51 \pm 0.05$ | $234.79 \pm 19.15$ |
| SCRIMP | $0.85 \pm 0.03$ | $77.86 \pm 11.09$ |
| VDN | $0.45 \pm 0.05$ | $242.42 \pm 17.92$ |

Table 4: Comparison of algorithms' cooperation on `Puzzles` set. $\pm$ shows confidence intervals 95%. Here, tan boxes highlight the best approach, and teal boxes highlight the best approach with a learnable part.

### B.5 PATHFINDING

To compute Pathfinding metric we run the approaches on the instances with a single agent. For this purpose we utilized large `Cities` maps with $256 \times 256$ size, the results are presented in Table 5. While this task seems easy, most of the hybrid and MARL approaches are not able to effectively solve them. Only LaCAM is able to find optimal paths in all the cases, as it utilizes precomputed costs to the goal location as a heuristic. Most of the evaluated hybrid and MARL approaches are also contain a sort of global guidance in one the channels of their observations. However, large maps with out-of-distribution structure, the absence of communication and other agents in local observations are able to lead to inconsistent behavior of the models that are not able to effectively choose the actions that lead the agent to its goal. Please note, SoC and makespan metrics in this case are equal, as there is only one agent in every instance.

| Algorithm | Makespan |
|---|---|
| DCC | $189.56 \pm 26.29$ |
| IQL | $1096.86 \pm 196.97$ |
| LaCAM | $179.82 \pm 20.97$ |
| MAMBA | $416.45 \pm 139.34$ |
| QMIX | $1055.75 \pm 193.03$ |
| QPLEX | $795.09 \pm 187.72$ |
| SCRIMP | $755.98 \pm 183.31$ |
| VDN | $1114.21 \pm 211.93$ |

Table 5: Comparison of makespan (the lower is better) used for pathfinding metric. tan boxes highlight the best approach, and teal boxes highlight the best approach with a learnable part.

### B.6 COORDINATION

The coordination metric is based on the number of collisions that occur during an episode. These collisions can occur either between agents, when two or more agents try to occupy the same cell or traverse the same edge simultaneously, or with static obstacles, when an agent tries to occupy a blocked cell. All such collisions are prevented by POGEMA during the action execution process, as all colliding actions are replaced with waiting actions instead. Figure 7 shows the average total

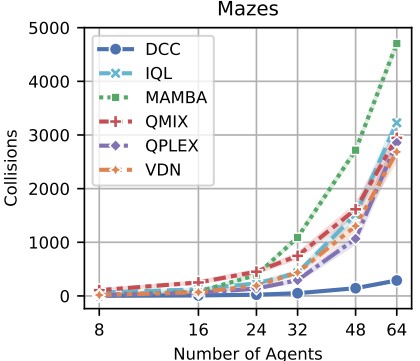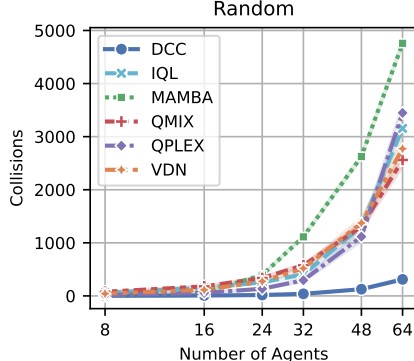

Figure 7: Total number of collisions occurred during solving MAPF instances with corresponding number of agents of `Mazes` and `Random` sets of maps.

number of collisions that occurred while solving instances with the corresponding number of agents on the `Mazes` and `Random` map sets.

The highest number of collisions occurred with the MAMBA approach, while the fewest were with DCC. The low number of collisions made by DCC compared to other approaches can be attributed to the presence of a communication mechanism that helps avoid collisions. The results of two evaluated approaches, LaCAM and SCRIMP, were omitted and are not presented in Figure 7. LaCAM is a centralized planner, so its solutions are collision-free by design. SCRIMP has an integrated environment with communication and tie-breaking mechanisms that resolve all collisions.

## C   RESULTS FOR LIFELONG MAPF BENCHMARK

In this section, we present the extended results of the LifeLong MAPF benchmark analysis, highlighting performance, out-of-distribution handling, scalability, cooperation, coordination, and pathfinding.

### C.1   PERFORMANCE

Performance metric in LMAPF case is based on the ratio of throughput compared to the best obtained one. In contrast to SoC, throughput should be as high as possible. There is also no CSR metric, as there is no need for agents to be at their goal locations simultaneously. As well as in MAPF case, the best results are obtained by centralized search-based approach – RHCR. The best results among decentralized methods demonstrate Follower and MATS-LP, following them, comparable results are shown by QPLEX, QMIX, ASwitcher which significantly outperforms MAMBA on both `Mazes` and `Random` maps. The results are presented in Figure 8.

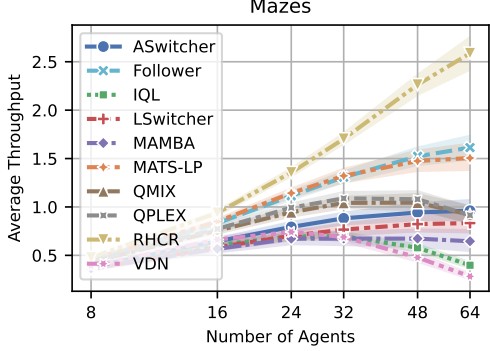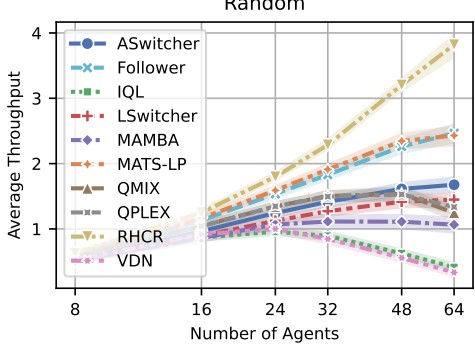

Figure 8: Performance results for LifeLong scenarios on the `Mazes` and `Random` maps.

## C.2   OUT-OF-DISTRIBUTION

The evaluation on out-of-distribution set of maps confirms the results obtained on `Random` and `Mazes` maps. The best results demonstrates RHCR. Next best results are obtained by Follower and MATS-LP, which are much closer to RHCR in this experiment. While MATS-LP outperforms Follower on the instances with 64, 128 and 192 agents, Follower is still better on the instances with 256 agents. Such relation is probably explained by the presence of dynamic edge-costs in Follower that allows to better distribute agents along the map and reduce coordination between them.

| Algorithm | 64 Agents | 128 Agents | 192 Agents | 256 Agents |
|---|---|---|---|---|
| ASwitcher | $1.26 \pm 0.08$ | $2.30 \pm 0.13$ | $3.14 \pm 0.17$ | $3.80 \pm 0.20$ |
| Follower | $1.50 \pm 0.08$ | $2.82 \pm 0.13$ | $3.95 \pm 0.19$ | $4.81 \pm 0.22$ |
| IQL | $1.10 \pm 0.06$ | $1.94 \pm 0.11$ | $2.32 \pm 0.15$ | $2.37 \pm 0.15$ |
| LSwitcher | $1.23 \pm 0.07$ | $2.23 \pm 0.12$ | $3.06 \pm 0.17$ | $3.67 \pm 0.20$ |
| MAMBA | $1.02 \pm 0.05$ | $1.42 \pm 0.08$ | $2.05 \pm 0.12$ | $2.46 \pm 0.17$ |
| MATS-LP | $1.57 \pm 0.12$ | $2.98 \pm 0.20$ | $4.04 \pm 0.33$ | $4.69 \pm 0.39$ |
| QMIX | $1.36 \pm 0.07$ | $2.54 \pm 0.12$ | $3.46 \pm 0.16$ | $4.03 \pm 0.20$ |
| QPLEX | $1.47 \pm 0.08$ | $2.67 \pm 0.12$ | $3.61 \pm 0.18$ | $4.22 \pm 0.22$ |
| RHCR | $1.57 \pm 0.08$ | $3.00 \pm 0.14$ | $4.22 \pm 0.23$ | $5.13 \pm 0.34$ |
| VDN | $1.12 \pm 0.06$ | $2.26 \pm 0.10$ | $2.81 \pm 0.14$ | $2.85 \pm 0.16$ |

Table 6: Evaluation on Out-of-Distribution maps. tan boxes highlight the best approach according to the average throughput metric, and teal boxes highlight the best approach with a learnable component.

## C.3   SCALABILITY

Figure 9 contains log-scaled plot of average time spent by each of the algorithms to process an instance on `Warehouse` map with the corresponding amount of agents. Most of the approaches scales almost linearly, except RHCR. This centralized search-based method lacks of exponential grow, as it needs to find a collision-free solution for at least next few steps, rather than just to make a decision about next action for each of the agents. The worst runtime demonstrate MATS-LP, as it runs MCTS and simulates the behavior of the other observable agents. It's still scales better than RHCR as it builds trees for each of the agents independently.

Figure 9: Runtime in seconds for each algorithm. Note that the plot is log-scaled.

## C.4   COOPERATION

As well as for MAPF setting, cooperation metric is computed based on the results obtained on `Puzzles` dataset. Table 7 contains average throughput obtained by each of the evaluated approaches. Here again the best results are obtained by RHCR algorithm. In contrast to `Random`, `Mazes` and `Warehouse` sets of maps, where MATS-LP and Follower demonstrate close results, the ability to simulate the behavior of other agents, provided by MCTS in MATS-LP, allows to significantly outperform Follower on small `Puzzles` maps. The rest approaches demonstrate much worse results, especially IQL, MAMBA, VDN that have almost 5 times worse average throughput than RHCR.

| Algorithm | Average Throughput |
|---|---|
| ASwitcher | $0.164 \pm 0.015$ |
| Follower | $0.319 \pm 0.020$ |
| IQL | $0.125 \pm 0.013$ |
| LSwitcher | $0.206 \pm 0.013$ |
| MAMBA | $0.133 \pm 0.014$ |
| MATS-LP | $0.394 \pm 0.021$ |
| QMIX | $0.228 \pm 0.018$ |
| QPLEX | $0.217 \pm 0.019$ |
| RHCR | $0.538 \pm 0.021$ |
| VDN | $0.144 \pm 0.015$ |

Table 7: Average throughput on `Puzzles` maps that were used to compute Cooperation metric.

### C.5 Pathfinding

Pathfinding metric is tailored to indicate how well the algorithm is able to guide an agent to its goal location. As a result, there is actually no need to run the algorithms on LifeLong instances. Instead, they were run on the same set of instances that were utilized for MAPF approaches.

The results of this evaluation are presented in Table 8. Again, the best results were obtained by search-based approach – RHCR. Its implementation was slightly modified to work on MAPF instances, when there is no new goal after reaching the current one. Either optimal or close to optimal paths are able to find MATS-LP, Follower and QPLEX. Followers misses optimal paths due to the integrated technique that changes the edge-costs. MATS-LP adds noise to the root of the search tree that might result in choosing of wrong actions. For approaches in the Switcher family, it is nearly impossible to find optimal paths, as they lack information about the global map and rely solely on local observations. Surprisingly, ASwitcher outperforms MAMBA, QMIX, IQL, and VDN, which are provided with a global map.

| Algorithm | Makespan |
|---|---|
| ASwitcher | $340.56 \pm 79.41$ |
| Follower | $181.00 \pm 20.95$ |
| IQL | $900.73 \pm 188.60$ |
| LSwitcher | $472.64 \pm 119.23$ |
| MAMBA | $416.45 \pm 136.01$ |
| MATS-LP | $179.93 \pm 22.45$ |
| QMIX | $461.90 \pm 147.16$ |
| QPLEX | $181.10 \pm 20.95$ |
| RHCR | $179.82 \pm 20.21$ |
| VDN | $1609.50 \pm 172.46$ |

Table 8: Pathfinding results.

### C.6 Coordination

Figure 10 illustrates the average total number of collisions that occurred during the solving of LMAPF instances. The absolute values are higher than those obtained during the solving of MAPF instances. This behavior is explained by the extended episode length in LMAPF instances, which is twice as long. Moreover, in MAPF scenarios, the episode can end when all agents reach their goal locations, whereas in LMAPF scenarios, all agents continue to act until the episode length limit is reached.

All MARL approaches show poor results, with MAMBA being the worst among them. The fewest collisions are made by the Switcher approaches, i.e., ASwitcher and LSwitcher. A comparable number of collisions is demonstrated by MATS-LP and Follower on the `Random` set of maps. The difference in the behavior of these two approaches on the `Random` and `Mazes` map sets is likely due to the more complex structure of obstacles on the `Mazes` maps, where their heuristic guidance more often leads to collisions. It's also worth noting that MATS-LP has no collisions with static obstacles, as it employs a masking mechanism that prevents selecting an action that leads an agent into a blocked cell. Such a mechanism could be implemented in other approaches to prevent this type of collision and potentially improve their performance. The results of the RHCR approach are omitted, as it is a centralized planner and its solutions are guaranteed to be collision-free.

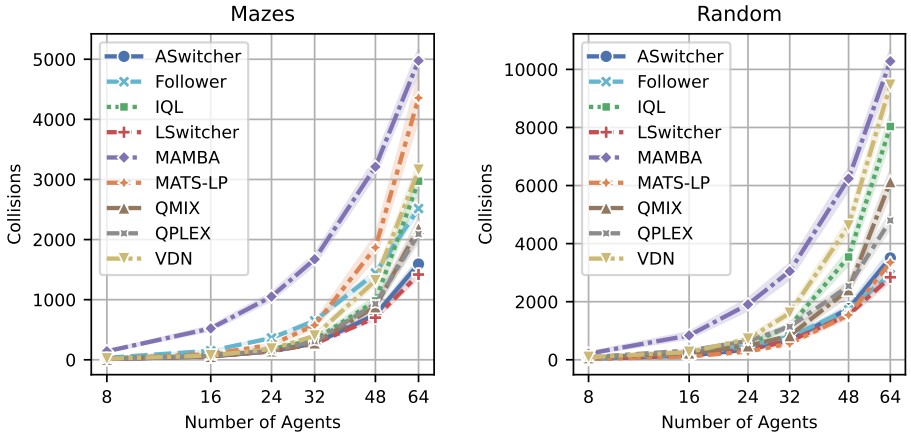

Figure 10: Total number of collisions occurred during solving LMAPF instances with corresponding number of agents of `Mazes` and `Random` sets of maps.

## D CODE EXAMPLES FOR POGEMA

POGEMA is an environment that provides a simple scheme for creating MAPF scenarios, specifying the parameters of `GridConfig`. The main parameters are: `on_target` (the behavior of an agent on the target, e.g., *restart* for LifeLong MAPF and *nothing* for classical MAPF), `seed` – to preserve the same generation of the map; agent; and their targets for scenario, `size` – used for cases without custom maps to specify the size of the map, `density` – the density of obstacles, `num_agents` – the number of agents, `obs_radius` – observation radius, `collision_system` – controls how conflicts are handled in the environment (we used a *soft* collision system for all of our experiments). The example of creation such instance is presented in Figure 11.

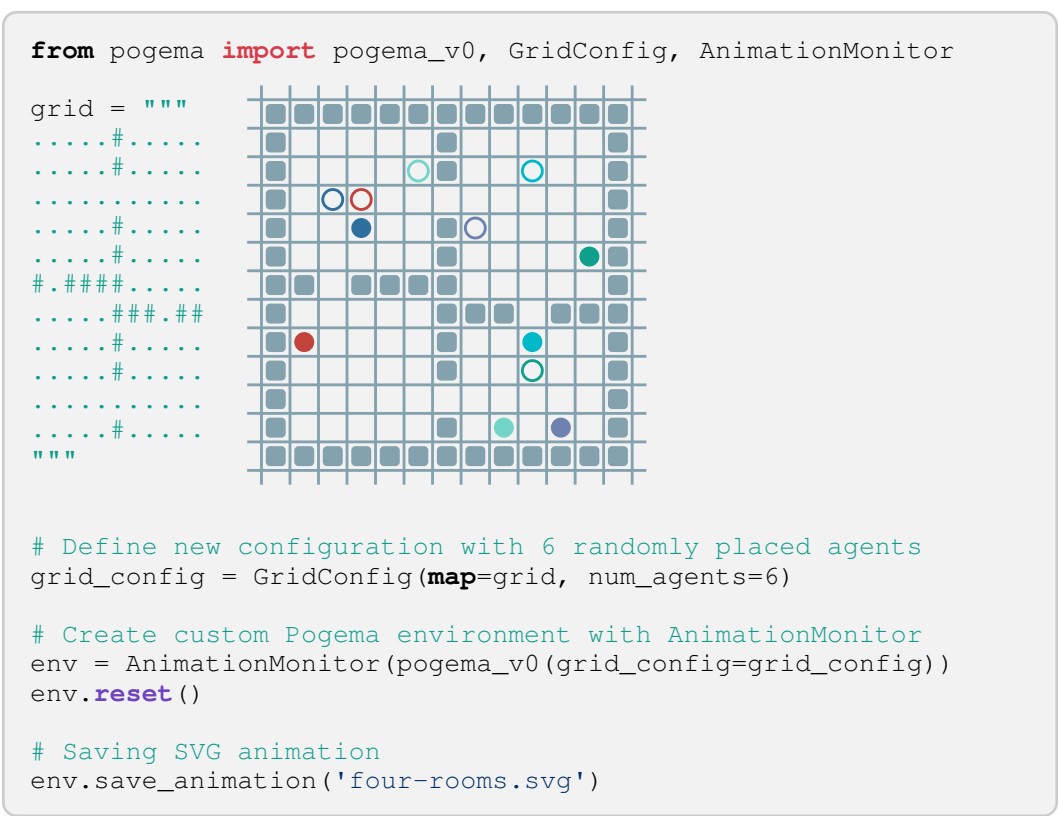

Figure 11: Setting up a POGEMA instance with a custom map and generating an animation.

Visualization of the agents is a crucial tool for debugging algorithms, visually comparing them, and presenting the results. Many existing MARL environments lack such tools, or have limited visualization functionality, e.g., requiring running the simulator to provide replays, or offering visualizations only in one format (such as videos). In the POGEMA environment, there are three types of visualization formats. The first one is console rendering, which can be used with the default `render` methods of the environment; this approach is useful for local or server-side debugging. The preferred second option is *SVG* animations. An example of generating such a visualization is presented in the listing above. This approach allows displaying the results using any modern web browser. It provides the ability to highlight high-quality static images (e.g., as the images provided in the paper) or to display results on a website (e. g., animations of the POGEMA repository on GitHub). This format ensures high-quality vector graphics. The third option is to render the results to video format, which is useful for presentations and videos.

# E  POGEMA TOOLBOX

The POGEMA Toolbox provides three types of functionality.

The first one is registries to handle custom maps and algorithms. To create a custom map, the user first needs to define it using ASCII symbols or by uploading it from a file, and then register it using the toolbox (see Figure 13). The same approach is used to register and create algorithms (see Figure 12). In that listing, the registration of a simple algorithm is presented, which must include two methods: `act` and `reset_states`. This approach can also accommodate a set of hyperparameters which the Toolbox handles.

```python
from pogema import BatchAStarAgent

# Registring A* algorithm
ToolboxRegistry.register_algorithm('A*', BatchAStarAgent)

# Creating algorithm
algo = ToolboxRegistry.create_algorithm("A*")
```

Figure 12: Example of registering the A* algorithm as an approach in the POGEMA Toolbox.

```python
from pogema_toolbox.registry import ToolboxRegistry

# Creating custom map
custom_map = """
.......#.
...#...#.
.#.###.#.
"""

# Registring custom map
ToolboxRegistry.register_maps({"custom_map": custom_map})
```

Figure 13: Example of registering a custom map to the POGEMA Toolbox.

Second, it provides a unified way of conducting distributed testing using Dask [16] and defined configurations. An example of such a configuration is provided in Figure 14. The configuration is split into three main sections; the first one details the parameters of the POGEMA environment used for testing. It also includes iteration over the number of agents, seeds, and names of the map (which were registered beforehand). The unified `grid_search` tag allows for the examination of any existing parameter of the environment. The second part of the configurations is a list of algorithms to be tested. Each algorithm has its alias (which will be shown in the results) and name, which specifies the family of methods. It also includes a list of hyperparameters common to different approaches, e.g., the number of processes, parallel backend, etc., and the specific parameters of the algorithm.

The third functionality and third part of the configuration concern views. This is a form of presenting the results of the algorithms. Working with complex testing often requires custom tools for creating visual materials such as plots and tables. The POGEMA toolbox provides such functionality for MAPF tasks out-of-the-box. The listing provides two examples of such data visualization: a plot and a table, which, based on the configuration, provide aggregations of results and present information in a high-quality form, including confidence intervals. The plots and tables in the paper are prepared using this functionality.

---

[16]https://github.com/dask/dask

```yaml
environment: # Configuring Test Environments
  name: Environment
  on_target: 'restart'
  max_episode_steps: 128
  observation_type: 'POMAPF'
  collision_system: 'soft'
  seed:
    grid_search: [ 0, 1, 2, 3, 4, 5, 6, 7, 8, 9 ]
  num_agents:
    grid_search: [ 8, 16, 24, 32, 48, 64 ]
  map_name:
    grid_search: [
        validation-mazes-000, validation-mazes-001,
        validation-mazes-002, validation-mazes-003,
        validation-mazes-004, validation-mazes-005,
    ]

algorithms: # Specifying algorithms and it's hyperparameters
  RHCR_5_10:
    name: RHCR
    parallel_backend: 'balanced_dask'
    num_process: 32
    simulation_window: 5
    planning_window: 10
    time_limit: 10
    low_level_planner: 'SIPP'
    solver: 'PBS'

results_views: # Defining results visualization
  MazesPlot:
    type: plot
    x: num_agents
    y: avg_throughput
    width: 4.0
    height: 3.1
    line_width: 2
    use_log_scale_x: True
    legend_font_size: 8
    font_size: 8
    name: Mazes
    ticks: [8, 16, 24, 32, 48, 64]

  TabularThroughput:
    type: tabular
    print_results: True
```

Figure 14: Example of the POGEMA Toolbox configuration for parallel testing of the RHCR approach and visualization of its results.

## F EXTENDED RELATED WORK

**StarCraft Multi-Agent Challenge** — The StarCraft Multi-Agent Challenge (SMAC) is a highly used benchmark in the MARL community. Most MARL papers that propose new algorithms provide evaluations in this environment. The environment offers a large set of possible tasks where a group of units tries to defeat another group of units controlled by a bot (a predefined programmed policy). Such tasks are partially observable and often require simple navigation. However, the benchmark has

several drawbacks, such as the need to use the slow simulator of the StarCraft II engine, deterministic tasks, and the lack of an evaluation protocol.

Nevertheless, some of these drawbacks have already been addressed. SMAX Rutherford et al. (2023) provides a hardware-accelerated JAX version of the environment, but it cannot guarantee full compatibility since the StarCraft II engine is proprietary software. SMAC v2 Ellis et al. (2024) solves the problem of determinism, highlighting this issue in the original SMAC environments. Moreover, an evaluation protocol for the SMAC environment is proposed in Gorsane et al. (2022). Despite these efforts, it's hard to say that these tasks require the generalization ability of the agent, since the training and evaluation are conducted on the same scenario.

**Multi-agent MuJoCo** — In MAMuJoCo, the standard tasks involve agents controlling different sets of joints (or a single joint) within a simulated robot. This set of environments is a natural adaptation of the environment presented in the well-known MuJoCo physics engine Todorov et al. (2012). These tasks don't require high generalization abilities or navigation. In the newer version, MuJoCo provides a hardware-accelerated version, forming the basis for Multi-agent BRAX Rutherford et al. (2023), which enhances performance and efficiency.

**Google Research Football** — Google Research Football Kurach et al. (2020) is a multi-agent football simulator that provides a framework for cooperative or competitive multi-agent tasks. Despite the large number of possible scenarios in the football academy and the requirement for simple navigation, the tasks are highly specific to the studied domain. Additionally, the number of possible agents is limited. Moreover, the framework offers low scalability, requiring a heavy engine.

**Multi-robot warehouse** — The multi-robot warehouse environment RWARE Papoudakis et al. (2021) simulates a warehouse with robots delivering requested goods. The environment is highly specific to delivery tasks; however, it doesn't support procedurally generated scenarios, thus not requiring generalization abilities or an evaluation protocol. The best-performing solution Christianos et al. (2020) in this environment is trained on only 4 agents. The benchmark is highly related to multi-agent pathfinding tasks; however, it doesn't provide centralized solution integration, which could serve both as an upper bound for learnable decentralized methods and as a source of expert demonstrations.

**Level-Based Foraging** — Multi-agent environment LBF Papoudakis et al. (2021) simulates food collection by several autonomously navigating agents in a grid world. Each agent is assigned a level. Food is also randomly scattered, each having a level on its own. The collection of food is successful only if the sum of the levels of the agents involved in loading is equal to or higher than the level of the food. The agents are getting rewarded by level of food they collected normalized by their level and overall food level of the episode. The game requires cooperation but also the agents can emerge competitive behavior. The environment is very efficiently designed and very simple to set up; however, it doesn't support procedurally generated scenarios, thus not requiring generalization abilities or an evaluation protocol.

**Flatland** — The Flatland environment Mohanty et al. (2020) is designed to address the specific problem of fast, conflict-free train scheduling on a fixed railway map. This environment was created for the Flatland Competition Laurent et al. (2021). The overall task is centralized with full observation; however, there is an adaptation to partial observability for RL agents. Unfortunately, during several competitions, despite the presence of stochastic events, centralized solutions Li et al. (2021a) from operations research field have outperformed RL solutions by a large margin in both quality and speed. The environment is procedurally generated, which requires high generalization abilities, and the benchmark provides an evaluation protocol. A significant drawback is the extremely slow speed of the environment, which highly restricts large-scale learning.

**Overcooked** — The Overcooked is a benchmark environment for fully cooperative human-AI task performance, based on the widely popular video game (Carroll et al., 2019). In the game, agents control chefs tasked to cook some dishes. Due to possible complexity of the cooking process, involving multistep decision-making, it requires emergence of cooperative behavior between the agents.

**Griddly** — This is a grid-based game engine (Bamford, 2021), allowing to make various and diverse grid-world scenarios. The environment is very performance efficient, being able to make thousands step per second. Moreover, there is test coverage and continuous integration support, allowing

open-source development. The engine provides support for different observation setups and maintains state history, making it useful for search based methods.

**Multi-player game simulators** — Despite the popularity of multi-player games, it's a challenging problem to develop simulators of the games that could be used for research purposes. One of the most popular adaptations are MineCraft MALMO (Johnson et al., 2016) that allows to utilize MineCraft as a configurable research platform for multi-agent research and model various agents' interactions. In spite of the game's flexible functionality, it depends on external runtime, so might be very hard to set up or extremely slow to iterate with. That's why there are several alternatives that prioritize fast iteration over the environment complexity, like Neural MMO Suarez et al. (2024) that models a simple MMO RPG with agents with a shared resource pool. On top of that, there are even faster implementation, targeting coordination or cooperation, like Hide-and-Seek Baker et al. (2020), which models competition, or GoBigger Zhang et al. (2023), focusing on competition between cooperating populations.

**Magent** —— Is a well-recognized environment within the community, though less widely used than SMAC. However, it lacks procedural generation capabilities, essential for testing agent generalization. The benchmark consists of six static scenarios, with the largest agent population observed in the *Gather* scenario, supporting up to 495 agents. The scenarios include *Adversarial Pursuit*, where red agents navigate obstacles to tag blue agents without causing damage; *Battle*, a large-scale team battle rewarding individual agent performance; *Battlefield*, similar to Battle but with fewer agents, but with predefined obstacles on the field; *Combined Arms*, a team battle featuring ranged and melee units with differing attack ranges, speeds, and health points; *Gather*, where agents compete for finite food resources requiring multiple "attacks" to consume; and *Tiger-Deer*, in which tigers team up to hunt deer, earning rewards through successful cooperation.

**Gigastep** — A GPU-accelerated multi-agent benchmark that supports both collaborative and adversarial tasks, continuous and discrete action spaces, and provides RGB image and feature vector observations. It features a diverse set of scenarios with heterogeneous agents, asymmetric teams, and varying objectives (e.g., tagging, waypoint following, hide-and-seek). However, it lacks procedural generation, which is crucial for testing generalization. Its evaluation protocol is minimal, relying on comparisons between PPO and hand-designed bots, offering limited insights into algorithmic diversity.

**Multi-agent Driving Simulators** — Autonomous driving is one of the important practical applications of MARL, and Nocturne Vinitsky et al. (2022) is a 2D simulator, written in C++, that focuses on different scenarios of interactions — e.g. intersections, roundabouts etc. The simulator is based on trajectories collected in real life, allowing it to model practical scenarios. This environment has evaluation protocols and supports open-source development with continuous integration and covered by tests. There are also environments, focusing on particular details of driving, for example, Multi-car Racing Schwarting et al. (2021) that represents racing from bird's eye view.

**Suits of multi-agent environments** — These multi-agent environments are designed to be very simple benchmarks for specific tasks. Jumanji Bonnet et al. (2023) is a set of environments for different multi-agent scenarios connected to combinatorial optimization and control, for example, routing or packing problems. With the purpose for each environment to be focused on the particular task, the overall suit doesn't test generalization or enable procedural generation. Multi Particle Environments (MPE) Lowe et al. (2017) is a communication oriented set of partially observable environments where particle agents are able to interact with fixed landmarks and each other, communicating with each other. SISL Gupta et al. (2017) is a set of three dense reward environments was developed to have a simple benchmark for various cooperative scenarios. For environment suits, testing generalization, MeltingPot Agapiou et al. (2022) comes into place. This set of the environments contains a diverse set of cooperative and general-sum partially observable games and maintains two populations of agents: focal (learning) and visiting (unknown to the environment) to benchmark generalization abilities of MARL algorithms. The set in based on the own game engine and might be extended quite easily.

**Real-world Engineering in Practice** — Real-world engineering tasks can often be addressed by MARL solutions. IMP-MARL Leroy et al. (2024) provides a platform for evaluating the scalability of cooperative MARL methods responsible for planning inspections and repairs for specific system components, with the goal of minimizing maintenance costs. At the same time, agents must cooperate

to minimize the overall risk of system failure. MATE Pan et al. (2022) addresses target coverage control challenges in real-world scenarios. It presents an asymmetric cooperative-competitive game featuring two groups of learning agents, cameras and targets, each with opposing goals.

## G    EXAMPLES OF USED MAPS

The examples of used maps are presented in Figure 15, showing a diverse list of maps. The map types used in the POGEMA Benchmark include: `Mazes`, with prolonged 1-cell width corridors requiring high-level cooperation; `Random`, easily generated maps to avoid overfitting with controllable obstacle density; `Cities-tiles`, smaller modified slices of `Cities` maps; `Puzzles`, small hand-crafted maps with challenging patterns necessitating agent cooperation; `Warehouse`, widely used in LifeLong MAPF research, featuring high agent density and throughput challenges; and `Cities`, large maps with varying structures for single-agent pathfinding.

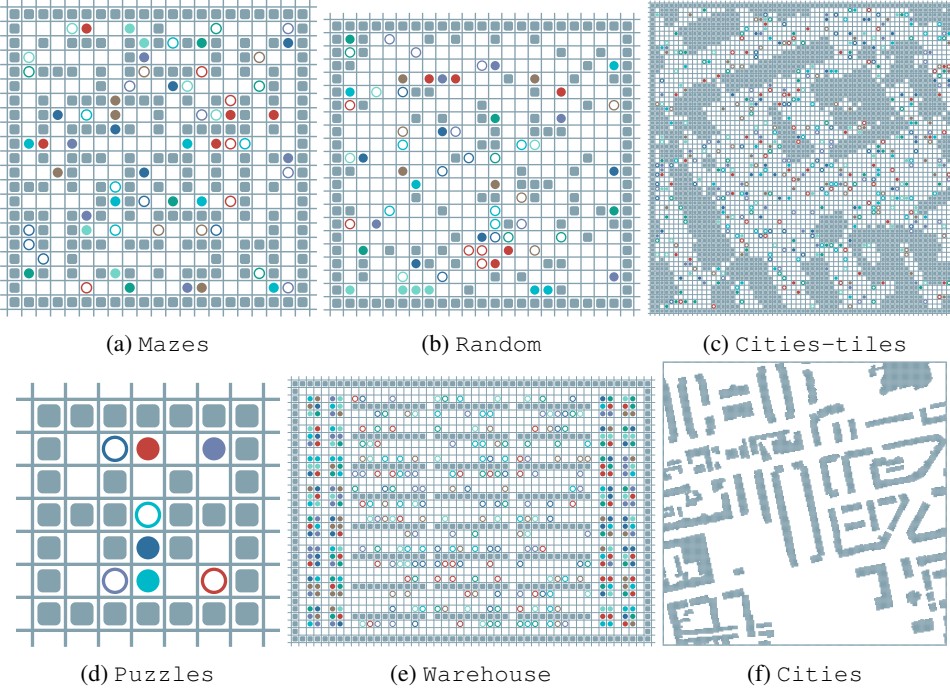

| (a) `Mazes` | (b) `Random` | (c) `Cities-tiles` |
| (d) `Puzzles` | (e) `Warehouse` | (f) `Cities` |

Figure 15: Examples of maps presented in the POGEMA Benchmark. The city map (on which the pathfinding metric was tested) is shown without grid lines and agents for clarity.

## H    MARL TRAINING SETUP

For training MARL approaches such as QMIX, QPLEX, IQL, and VDN, we started based on the default hyperparameters provided in the corresponding repositories, employing the PyMARL2 framework[17]. These hyperparameters are mostly tuned for the SMAC environment, so we tuned

| Hyperparameter | IQL | QPLEX | QMIX | VDN |
|---|---|---|---|---|
| Batch Size | 64 | 64 | 32 | 32 |
| Learning Rate | 0.002 | 0.002 | 0.001 | 0.001 |
| RNN Size | 128 | 128 | 256 | 256 |

Table 9: Best hyperparameters found by hyperparameter sweep, which is different from defaults ones.

the main ones for our use case. For this hyperparameter sweep, we used grid search over parameters such as learning rate, batch size, replay buffer size, and neural network parameters like the size of RNN blocks. We used the default functionality of the Wandb framework[18] for this sweep, with the optimization target being the CSR of the agent on the training maps. The best found hyperparameters

---

[17] https://github.com/hijkzzz/pymarl2
[18] https://github.com/wandb/wandb

which is different from default ones are presented in Table 9. We used the default hyperparameters for MAMBA, provided in corresponding repository[19].

As input, we apply preprocessing from the Follower approach, which is the current state-of-the-art for decentralized LifeLong MAPF. We attempted to add a ResNet encoder, as used in the Follower approach; however, this addition worsened the results, thus we opted for vectorized observation and default MLP architectures. For centralized methods that work with the state of the environment (e.g., QMIX or QPLEX), we utilized the MARL integration of POGEMA, which provides agent positions, targets, and obstacle positions in a format similar to the SMAC environment (providing their coordinates).

Our initial experiments on training this approach with a large number of agents, similar to the Follower model, showed very low results. We adjusted the training maps to be approximately $16 \times 16$, which proved to be more effective and populated them with 8 agents. All the MARL approaches were trained using the `Mazes` map generator. This setup produced better results. We continued training the approaches until they reached a plateau, which for most algorithms is under 1 million steps.

## I    RESOURCES AND STATISTICS

To evaluate all the presented approaches integrated with POGEMA we have used two workstations with equal configuration, that includes 2 NVidia Titan V GPU, AMD Ryzen Threadripper 3970X CPU and 256 GB RAM. The required computation time is heavily depends on the approach by itself.

|  | Random | Mazes | Warehouse | Cities-tiles | Puzzles | Cities |
|---|---|---|---|---|---|---|
| DCC | 2.11 | 2.46 | 11.07 | 22.70 | 0.09 | 0.02 |
| IQL | 0.05 | 0.04 | 0.13 | 0.13 | 0.01 | 0.01 |
| LaCAM | 0.20 | 0.29 | 0.24 | 0.23 | 0.37 | 0.01 |
| MAMBA | 6.62 | 6.47 | 8.36 | 12.27 | 2.59 | 3.40 |
| QMIX | 0.04 | 0.04 | 0.14 | 0.13 | 0.01 | 0.01 |
| QPLEX | 0.05 | 0.04 | 0.13 | 0.13 | 0.01 | 0.01 |
| SCRIMP | 1.66 | 2.20 | 16.54 | 21.63 | 0.08 | 0.21 |
| VDN | 0.05 | 0.04 | 0.13 | 0.13 | 0.01 | 0.01 |

Table 10: Total time (in hours) required by each of the algorithms to run all MAPF instances on the corresponding datasets.

|  | Random | Mazes | Warehouse | Cities-tiles | Puzzles | Cities |
|---|---|---|---|---|---|---|
| ASwitcher | 1.03 | 0.47 | 2.95 | 1.76 | 0.31 | 0.04 |
| Follower | 0.48 | 0.23 | 0.69 | 0.77 | 0.26 | 0.89 |
| IQL | 0.08 | 0.04 | 0.26 | 0.24 | 0.02 | 0.01 |
| LSwitcher | 6.18 | 2.61 | 17.30 | 10.70 | 0.81 | 0.21 |
| MAMBA | 13.82 | 6.69 | 15.81 | 11.07 | 7.83 | 3.40 |
| MATS-LP | 77.31 | 35.34 | 163.68 | 129.78 | 3.80 | 0.14 |
| QMIX | 0.08 | 0.04 | 0.26 | 0.25 | 0.02 | 0.01 |
| QPLEX | 0.08 | 0.04 | 0.26 | 0.25 | 0.02 | 0.01 |
| RHCR | 0.57 | 0.25 | 17.04 | 6.28 | 0.01 | 0.01 |
| VDN | 0.08 | 0.04 | 0.25 | 0.25 | 0.02 | 0.01 |

Table 11: Total time (in hours) required by each of the algorithms to run all LMAPF instances on the corresponding datasets.

The statistics regarding the spent time on solving MAPF and LMAPF instances are presented in Table 10 and Table 11 respectively. Please note, that all these approaches were run in parallel in multiple threads utilizing Dask, that significantly reduces the factual spent time.

We used pretrained models for all the hybrid methods, such as Follower, Switcher, MATS-LP, SCRIMP, and DCC, thus, no resources were spent on their training. RHCR and LaCAM are pure

---

[19]https://github.com/jbr-ai-labs/mamba

search-based planners and do not require any training. MARL methods, such as MAMBA, QPLEX, QMIX, IQL, and VDN, were trained by us. MAMBA was trained for 20 hours on the MAPF instances, resulting in 200K environment steps, and for 6 days on LifeLong MAPF instances, resulting in 50K environment steps, which corresponds to the same amount of GPU hours. For MARL approaches, we trained them for 1 million environment steps, which corresponds to an average of 5 GPU hours for each algorithm.

## J    COMMUNITY ENGAGEMENT AND FRAMEWORK ENHANCEMENTS

Our team is committed to maintaining an open and accountable POGEMA framework. We ensure transparency in our operations and encourage the broader AI community to participate. Our framework includes a fast learning environment, problem instance generator, visualization toolkit, and automated benchmarking tools, all guided by a clear evaluation protocol. We have also implemented and evaluated multiple strong baselines that simplify further comparison. We practice rigorous software testing and conduct regular code reviews.

## K    INGESTION OF MOVINGAI MAPS

We incorporated an ingestion script to convert MovingAI maps to be compatible with POGEMA. The ingestion script is straightforward, and an example of its usage is presented in Figure 16. This script downloads the full archive of, in this case, the street-map series, which will be saved to a YAML file compatible with POGEMA.

```python
from pogema_toolbox.generators.generator_utils import (
    maps_dict_to_yaml)
from pogema_toolbox.moving_ai_ingestion import (
    download_moving_ai_maps)

url = 'https://movingai.com/benchmarks/street/street-map.zip'
maps = download_moving_ai_maps(url)
maps_dict_to_yaml('maps.yaml', maps)
```

Figure 16: Ingestion script to convert MovingAI maps into POGEMA-compatible format

Instead of converting and distributing these maps ourselves, we provide this script to allow users to convert the maps on their own, ensuring compliance with licensing terms. Additionally, by using this script, users can always work with the most up-to-date versions of the maps in the MovingAI dataset, addressing any changes or updates made to the dataset over time.

## L    POGEMA SPEED PERFORMANCE EVALUATION

The speed of an environment is a critical aspect in reinforcement learning (RL), significantly influencing training performance and usability. Table 1 provides an overview of the speed performance of various multi-agent environments, demonstrating that POGEMA can process more than 10K steps per second.

Here we provide more detailed information about POGEMA's speed performance. POGEMA's continuous integration (CI) includes a speed measurement procedure that runs alongside the tests. Here, we present the results from a recent CI run. We compared POGEMA's performance on three CPU setups: the AMD Ryzen Threadripper 3970X 32-Core Processor, a modern high-performance CPU; an older server-side Intel(R) Xeon(R) CPU @ 2.20GHz, commonly used on the Google Colab platform, representing an average-performance setup; and the Apple Silicon M1, a laptop setup that is widely regarded for its power efficiency and solid performance in computational tasks.

For this test, we used default observation parameters commonly employed in learnable multi-agent pathfinding (MAPF) approaches (e.g., Follower, SCRIMP, DCC). Specifically, we set the observation

| Agents | Size | Ryzen Threadripper | | Intel(R) 2.20GHz | | Apple Silicon M1 | |
|---|---|---|---|---|---|---|---|
| | | MAPF | LMAPF | MAPF | LMAPF | MAPF | LMAPF |
| 1 | 32 | 20 684 | 21 810 | 9391 | 13 358 | 29 699 | 44 224 |
| 1 | 64 | 10 390 | 9602 | 6981 | 6452 | 19 244 | 18 996 |
| 32 | 32 | 96 918 | 90 631 | 61 132 | 61 232 | 204 637 | 191 069 |
| 32 | 64 | 89 984 | 85 741 | 61 297 | 38 326 | 144 125 | 175 962 |
| 64 | 32 | 111 976 | 105 558 | 69 121 | 39 308 | 189 624 | 217 482 |
| 64 | 64 | 102 104 | 96 709 | 68 126 | 49 902 | 196 808 | 194 213 |

Table 12: POGEMA performance in observations per second (OPS) across different CPU types using a single CPU core. Note that the reported OPS represents the total frames (observations) received by all agents, not the environment steps.

radius to 5, corresponding to an $11 \times 11$ field. The results for both MAPF and LMAPF scenarios, using a random policy, are reported in Table 12. In scenarios with more than 32 agents, POGEMA achieves $\geq 80K$ steps per second (OPS) on fast CPUs like the Ryzen Threadripper and Apple Silicon M1, and $\geq 38K$ OPS on the Intel Xeon setup, which is notably fast for a single-environment, single-thread configuration. For comparison, EnvPool[20] reports 50K frames per second (FPS) for Atari on a 12-CPU setup.

We further investigate POGEMA's speed performance using SampleFactory[21] as the sampler for parallel asynchronous execution of the environment. We used an almost default configuration with a random policy (instead of PPO) and employed two environments per worker (for double buffering in SampleFactory). The number of sampling workers varied alongside the number of agents in the environment, and the tests were run on a workstation equipped with a single AMD Ryzen Threadripper 3970X 32-Core Processor (64 threads). We conducted a sampling procedure for 5 minutes for each setup using `Random` maps with a size of $32 \times 32$. The results both MAPF and Lifelong MAPF scenarios are presented in Figure 17.

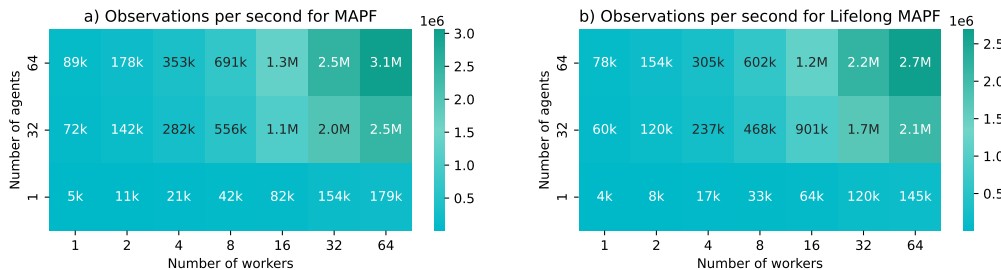

Figure 17: Observations per second performance of POGEMA for (a) MAPF and (b) Lifelong MAPF across different number of workers and agents in the environment, using AMD Ryzen Threadripper 3970X 32-Core Processor (64 threads). We tested `Random` maps with size $32 \times 32$.

Looking at the single-worker setup, it is notable that the performance from the previous experiment with a single CPU setup is noticeably higher. This slowdown can be attributed to the overhead produced by the parallel asynchronous execution of the framework. However, it allows to significantly improve performance; the results for the best configuration with 64 agents and 64 workers achieved 3.1M OPS for MAPF and 2.7M OPS for Lifelong MAPF. For both setups, only 16 workers are needed to exceed the significant threshold of 1M OPS. The best performance is observed in the MAPF scenario, where 916,464,000 samples were generated in 5 minutes. This demonstrates the efficiency of utilizing a high number of CPU cores for large-scale sampling tasks, which scaled almost linearly up to 32 workers. This also indicates that performance can be further improved with additional CPU resources.

---

[20]https://github.com/sail-sg/envpool
[21]https://github.com/alex-petrenko/sample-factory

To compare, we can refer to the JaxMARL paper, which provides insights into the speed performance of XLA-accelerated environments. The repository includes several environments, and we will focus on the SPS of the STORM environment, which offers grid-based tasks. Based on the paper, the speed of the environment is 2.48k with a single environment, 175k for 100 environments, and 14.6M SPS for 10,000 environments. These results were obtained using a single NVIDIA A100 GPU.

While, as expected, POGEMA is slower in very large vectorized setups compared to XLA vectorized environments that use GPU or TPU hardware acceleration, this trade-off has its advantages. It is challenging to devise an approach that can effectively utilize such a large amount of data. Additionally, by not relying on GPUs or TPUs for environment simulation, these resources remain fully available for training neural networks, which often represent the primary bottleneck in large-scale RL experiments.

We further evaluate the scalability of POGEMA with a large population of agents. For this purpose, we use a random map scenario of size $3072 \times 3072$ and test agent counts starting from 1,000 agents. The results, presented in Table 13, demonstrate that POGEMA efficiently supports up to 1 million agents within a single environment.

| Agents | OPS | SPS | Reset (seconds) |
|--------|------|------|-----------------|
| 1,000,000 | 68700.6 | 0.0687 | 173.8 |
| 100,000 | 67104.9 | 0.6710 | 139.6 |
| 10,000 | 45894.7 | 4.589 | 132.8 |
| 1,000 | 56477.7 | 56.477 | 120.6 |

Table 13: Speed performance of POGEMA with large agent populations using a random policy.

In previous experiments, we relied on a random policy to evaluate the environment's speed performance, as it is a common choice for such tests. However, the speed performance of the environment may vary when a more advanced approach is used, as it explores a larger portion of the state space compared to the random policy. To better test the speed performance of POGEMA under real inference, we used the Follower approach and LifeLong Mazes scenarios, with a size of $128 \times 128$ and up to 2048 agents, as they required creating new goals for agents upon reaching them. The results are presented in Figure 18. Here, one can see that with an increasing number of agents, the SPS of POGEMA decreases almost linearly. Additionally, the OPS throughput grows with the number of agents. This setup also highlights the ability of POGEMA to handle a large population of agents operating in the same environment. The experiment was conducted on a setup with an AMD Ryzen Threadripper 3970X 32-core processor, using a single CPU core and a single environment (no parallelization).

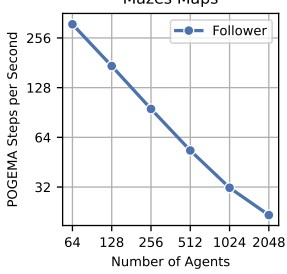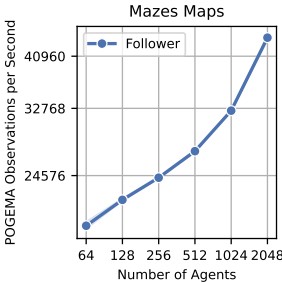

Figure 18: Steps per second and observations per second throughput of POGEMA with a large agent population using the Follower approach (without parallelization). Both axes are on a log scale.

