# OpenReview forum: "POGEMA: A Benchmark Platform for Cooperative Multi-Agent Pathfinding"
_ICLR.cc/2025/Conference — ICLR 2025 Poster_

### Official Review · Reviewer_Jpm1 · 2024-11-03

**Soundness:** 3
**Presentation:** 4
**Contribution:** 2
**Rating:** 6
**Confidence:** 4

**Summary:**

The paper proposes a new benchmark for multi-agent path-planning and reinforcement learning. The main contribution of the new benchmark appears to be its ability to be computationally efficient, the ability to procedurally generate new problems, and to allow a diverse set of map styles. The tasks are generally navigation tasks, in which an agent has to avoid collisions and reach a goal. The paper presents sufficient details on the new benchmark and compares a large number of different MARL and other multi-agent path planning algorithms.

**Strengths:**

- I like the extensive comparison and differentiation to previous multi-agent environments
- I appreciate the large number of evaluated algorithms on the new benchmark
- The analysis and experimental studies in this paper are strong
- The new environment seems to be in particularly interesting when evaluating a large number of agents in large environments
- I like that the authors promise to release the code under an open-access license.

**Weaknesses:**

- While I appreciate the computational efficiency and the ability to procedurally generate new environments, I am wondering after reading the paper if the benchmark can further the field by sparking new ideas or raising open problems. The benchmark seems more like an engineering achievement allowing the study of larger agent populations, but I am wondering if this is really _the_ open problem we need to consider in MARL.
- Similarly, the observation space seems relatively simple. I may have missed it, but there is also no uncertainty or noise in the sensor measurements. Again, this reinforces the impression that the introduced task-sets are relatively simple, I am failing a bit to see that this will truly push forward the field.
- This is due to the fact that the considered environments are, in principle, simple, as they are _just_ navigation tasks requiring a minimalistic representation. Again, I worry this is simply a too-simple set of tasks to further the research frontier...

**Questions:**

Feel free to respond to my concerns raised in the weaknesses. Overall, I think the implementation work seems good, the evaluation of algorithms extensive, but I fail to see the truly novel aspect where this will raise new open questions for multi-agent planning, new avenues for research or challenge the existing algorithms.

Hence, a rating of 6 may be the highest I am willing to give the paper in its current state.

---

> ### Author Response · Authors · 2024-11-23
>
> W1: We believe that achieving strong performance in decentralized settings, or even matching the scores of centralized methods, represents a significant breakthrough. The benchmark highlights this challenge and provides a platform to address it. Additionally, many MARL approaches underperform compared to learnable MAPF methods, even when using preprocessing tecniques from learnble MAPF (e.g. from Follower). This indicates that POGEMA presents a compelling challenge for large-scale MARL training setups, especially for the CTDE (centralized training decentralized execution) paradigm.
>
> We also want to emphasize several promising research directions in decision-making that can be explored using our benchmark:
> 1. Large-scale imitation learning:
> POGEMA provides a fast environment and efficient centralized baselines that can be used to generate high-quality data for training. This is particularly useful for foundation models, as the procedural generation of maps allows for an unlimited supply of expert data.
> 2. Communication learning:
> POGEMA’s large maps inherently require agents to rely on local communication, making it an ideal testbed for MARL approaches focusing on communication. While communication has been extensively studied in MARL, its application in large-scale settings remains underexplored, and POGEMA provides a platform for advancing this field.
> 3. Memory-efficient methods:
> Most existing approaches (except for Switcher) rely on global guidance to reach goals in POGEMA, which poses significant challenges in multi-agent scenarios requiring memory. The sharp drop in performance compared to  methods, which uses target guidance, highlights the need for memory-efficient strategies, making this an important area for further exploration.
> 4. Heterogeneous policies and opponent / ally modelling
> Currently, no centralized approaches can effectively handle scenarios with multiple policies. Learnable methods have the potential to close this gap, enabling agents to coordinate effectively in heterogeneous settings. This also opens an exciting avenue for studying how different algorithms can be trained concurrently in the same environment, fostering advancements in collaborative and adversarial multi-agent interactions.
>
> W2: The primary challenge in any multi-agent navigation problem is coordinating the move/wait actions of agents to balance safety (avoiding collisions) with efficiency (minimizing the time to reach goals). While it is true that practical scenarios often include additional complexities, such as uncertainties in observations or execution, even under the simplified assumptions of perfect sensing and execution, the problem remains extremely difficult—especially when dealing with large groups of agents and complex map topologies.
>
> Studying such “idealized” problems allows researchers to focus on the core challenges of coordination and cooperation without the confounding effects of noise or uncertainty. Insights gained from solving these problems can then be extended to more realistic scenarios, thereby advancing the field.
>
> W3: Even in idealized, fully observable, discretized, and fully synchronized setups, obtaining an optimal solution to multi-agent pathfinding is an NP-hard problem. Much of the research in this field focuses on striking the right balance between the solver’s speed and the quality of the solution—a task that remains far from simple. This trade-off is crucial for practical applications and continues to drive progress in the field.
>
> One promising approach to achieving this balance lies in decentralized, learnable methods, which POGEMA is specifically designed to support. By focusing on tasks with minimalistic representations, researchers can isolate and address fundamental challenges, enabling progress that can later extend to more complex and realistic scenarios. Thus, we believe that focusing on tasks with minimalistic representation one can actually push the frontiers forward.

---

> > ### Comment · Reviewer_Jpm1 · 2024-12-01
> >
> > I thank the authors for their detailled response. Overall, my view of the paper remains, as was also mentioned by other reviewers: While you point out that there may be potential research directions to be exploitet here, it remains for me too unclear and not well-defined enough, that I can believe that these objectives cannot be investigated with the currently existing MARL benchmarks.
> >
> > What the paper is really missing is some insight or research contribution from which it is clear that these results can only be obtained using your benchmark, and where a similar experiment may not be feasible with an already existing benchmark or not lead to the same conclusions. To be clear, I would not expect you to run these experiments against many other benchmarks, but at least a sensible selection.
> >
> > To me the main benefit you promise is the vast number of agents the environment can simulate - maybe the best approach would be hence to maintain the same computational budget for algorithms across different benchmarks, and show that because you allow RL or other leanring-based methods to maintain a larger population of agents that there are better generalization effects. But this would require some minimal benchmark-to-benchmark comparison.
> >
> > That said, these are the criteria I would be looking for to raise my score to 8. However, the paper is overall in a good state, and together with the extensive experimental study it is well above the threshold for acceptance - hence I do maintain my score of 6.

---

> > > ### Author Response · Authors · 2024-12-03
> > >
> > > We sincerely thank the reviewer for their thoughtful feedback and for acknowledging the extensive experimental study and overall good state of our paper. We greatly appreciate their willingness not only to provide a score but also to suggest valuable directions for improvement.
> > >
> > > During the initial review process, as per reviewer mDvH’s request regarding our claim of supporting >1000 agents in POGEMA, we recognized that this claim was not directly substantiated in the paper. This led us to investigate the maximum number of agents POGEMA can simulate. For context, to the best of our knowledge, classical MAPF approaches, such as PIBT[1], have demonstrated pathfinding for up to 10,000 agents in the same environment. Building on this, we explored whether POGEMA could support even larger numbers of agents and evaluated learning-based MAPF methods at these scales.
> > >
> > > We are pleased to report that POGEMA successfully supports up to 1 million agents in a single environment.
> > >  Below are summarized results on the speed performance of the environment (3072×3072, we used random policy to receive these results):
> > >
> > > | Number of Agents | Observations per Second | Steps per Second | Reset Runtime (seconds)  |
> > > |------------------|-------------------------|------------------|--------------------------|
> > > | 1,000,000        | 68700.6                 | 0.0687           | 173.8                    |
> > > | 100,000          | 67104.9                 | 0.6710           | 139.6                    |
> > > | 10,000           | 45894.7                 | 4.589            | 132.8                    |
> > > | 1,000            | 56477.7                 | 56.477           | 120.6                    |
> > >
> > >
> > > Beyond evaluating random policies, we also tested a recent state-of-the-art approach, MAPF-GPT [2], which leverages large-scale imitation learning policies for MAPF using POGEMA to generate demonstrations.
> > > Below are the results of inference for MAPF-GPT (2M parameters) on 1 million agents:
> > >
> > > |             | Average Throughput | Observations per second | Full experiment runtime |
> > > |-------------|--------------------|-------------------------|-------------------------|
> > > | MAPF-GPT-2M | 70.6               | 747.3                   | ~48 hours               |
> > >
> > >
> > > In addition to advancing MAPF, POGEMA is also a valuable tool for the MARL community. It highlights critical challenges for scaling MARL algorithms. For instance, the vanilla CTDE paradigm is impractical at this scale due to the infeasibility of encoding a full state or approximating value functions effectively. As noted, scaling MARL requires novel approaches. Popular methods such as QMIX and MAMBA struggle to handle large populations in such setups, further emphasizing the need for innovation in this domain.
> > >
> > > We hope this contribution helps address the reviewer’s concern regarding the unique insights and research contributions provided by our benchmark. The demonstrated scalability and ability to test learning-based methods at unprecedented scales highlight why these results are uniquely achievable using POGEMA. We will update the comparison table in the paper to reflect this improvement, replacing the claim of >1000 agents with ≥1 million agents, where our benchmark remains the only one capable of achieving this.
> > >
> > > [1] Okumura K, Machida M, Défago X, Tamura Y. Priority inheritance with backtracking for iterative multi-agent path finding. Artificial Intelligence. 2022 Sep 1;310:103752.
> > >
> > > [2] Andreychuk A, Yakovlev K, Panov A, Skrynnik A. MAPF-GPT: Imitation learning for multi-agent pathfinding at scale. arXiv preprint arXiv:2409.00134. 2024 Aug 29.

---

### Official Review · Reviewer_iVSi · 2024-11-04

**Soundness:** 2
**Presentation:** 2
**Contribution:** 3
**Rating:** 5
**Confidence:** 4

**Summary:**

This paper presents a benchmark for multi-agent navigation tasks. The overall work includes a fast and scalable environment for multi-agent navigation tasks, a series of built-in tasks, a set of visualization tools, and an evaluation framework.

**Strengths:**

1. The overall platform proposed in this paper is capable of integrating search-based, learning-based, and hybrid approaches together, using the same metric for comparison.
2. As an environment that supports large-scale multi-agent reinforcement learning algorithm training, performing more than 10K steps per second is quite remarkable.

**Weaknesses:**

(Major) The primary issue with this paper is that its contributions are not sufficiently prominent. As a benchmark in the multi-agent systems domain, POGEMA's positioning is unclear and lacks irreplaceable advantages. For instance, for a researcher in the MARL domain, it may mainly offer features related to partial observability, rapid training, and scalability to >1000 agents. However, benchmarks already exist in the MARL domain that possess these characteristics, such as [1][2], making this work not irreplaceable. For a researcher interested in the domain of multi-agent cooperative navigation, the overall scenario elements of this work are overly simplified and idealized, presenting a significant gap from real-world scenarios. It might be better to focus more on issues in real-world applications, such as unexpected situations in open environments and the transferability of algorithms from simulation to reality.


(Minor) There are concerns and suggestions regarding the writing of the introduction part. For example, the content about decentralized cooperative learning discussed in lines 53-58, is it closely related to the necessity of this work? Similarly, the transition in lines 59-77 related to partial observability and large-scale agents is too lengthy and seems not to be the focus of the problems the paper intends to solve. A well-written introduction should quickly highlight contradictions and then state the necessity of the work. Writing too much content with no clear relevance can impact readability.

[1] Lechner M, Seyde T, Wang T H J, et al. Gigastep-one billion steps per second multi-agent reinforcement learning[J]. Advances in Neural Information Processing Systems, 2024, 36.

[2] Zheng L, Yang J, Cai H, et al. Magent: A many-agent reinforcement learning platform for artificial collective intelligence[C]//Proceedings of the AAAI conference on artificial intelligence. 2018, 32(1).

**Questions:**

In addition to my primary concerns, there are also the following questions:
1. Please carefully check the writing of the paper. For example, on the first line of the third page, "both" needs to be removed.
2. Is it "NP-Hard" or "HP-Hard" in line 88 of the paper?
3. The definitions of competitive and cooperative behaviors in lines 221-225 might not be rigorous enough. For instance, in competitive behaviors, the rewards of agents might not sum up to a fixed value but could have a relationship akin to being inversely proportional; in cooperative behaviors, agents do not necessarily share rewards, but they are working towards the same task. Do the authors have a more rigorous perspective on this?

---

> ### Author Response · Authors · 2024-11-21
>
> We sincerely thank the reviewer for their time and expertise.
>
> W1: First, we want to emphasize that our primary contribution is POGEMA as a benchmark, not just as an environment. There are two main problems in MARL: reproducible evaluation and the generalization problem. The POGEMA benchmark addresses these by providing an evaluation protocol and tools, metrics, along with well-prepared baselines, including centralized MAPF solvers. Additionally, it includes procedural generation tools, which require agents to learn generalized policies that are then tested on unseen scenarios in a hold-out set during training.
>
> We thank the reviewer for pointing out papers [1][2], and we have included a comparison with them in Table 1. Let’s now review these environments.
>
> |Environment|Python Based |Hardware-Agnostic Setup |Procedural Generation |Requires Generalization |Evaluation Protocols |Tests & CI |>1000 Agents |
> |---|---|---|---|---|---|---|---|
> |POGEMA |✓|✓|✓ |✓|✓|✓|✓ |
> |Magent |✗|✗|✗ |✗|✗|✗|✗ |
> |Gigastep |✗|✗|✗ |✗|✗|✗|✓ |
>
>
> In comparison to Gigastep, we highlight several key distinctions:
> - POGEMA supports procedural generation of scenarios, which is essential for testing generalization. In contrast, Gigastep primarily uses empty maps or simple, hand-designed obstacles that do not challenge the diversity of algorithmic capabilities.
> - A vital characteristic of a benchmark is a well-defined evaluation protocol for assessing and comparing solutions. Gigastep lacks this, as it only compares PPO with hand-designed bots.
> - While GPU acceleration is a key feature of Gigastep, we argue it may be unnecessary. Once environments can process a sufficient number of steps per second across parallel environments, the bottleneck often shifts from simulation to neural network computation. In contrast, POGEMA’s approach—avoiding reliance on GPUs or TPUs for simulation—keeps these resources available for training. Additionally, we extended the Speed Performance Evaluation section in the Appendix, where we compared POGEMA to JaXMARL. POGEMA also outperforms Gigastep in reported throughput, generating a maximum of 3.1M observations per second on CPU.
> - Gigastep’s GitHub repository has not been actively maintained since 2022, with several unresolved issues, raising concerns about its long-term reproducibility. In contrast, POGEMA is actively maintained. We provided [version updates history](https://anonymous.4open.science/r/pogema-7439/version_history.MD) to anonymized code to support this claim.
>
> Magent is a well-recognized environment within the community, though less widely used than SMAC. However, it lacks procedural generation capabilities, essential for testing agent generalization. The benchmark consists of six static scenarios, none of which scale to more than 1,000 agents, with the largest agent population observed in the “Gather” scenario, supporting up to 495 agents.
>
> Both Gigastep and Magent require knowledge of JAX or C++ for modification, which can hinder their adaptability. In contrast, POGEMA is implemented in pure Python, making it more accessible and easier to modify.
>
> Next, we wish to elaborate on the need to address real-world complications in a multi-agent navigation environment. We agree with the reviewer that real-world multi-agent problems impose numerous challenges (continuous workspace and time, non-syncronized time, communication and observation issues, perception limitations etc.). Still, the core challenge of any multi-agent navigation problem is the coordination of actions between the agents in a way that minimises the risk of collisions and optimises a given cost objective. This aspect is the crux of any multi-agent navigation scenario and that is why we opted to distil this problem and focus on it in POGEMA. It is known that even the discretized version of the considered multi-agent pathfinding (MAPF) problem is NP-Hard to solve optimally. On the other hand, numerous practically important applications mimic the discrete nature of MAPF and actively use MAPF solutions in real world. The most prominent example being the automated warehouses where, indeed, robots move synchronously utilizing cardinal moves. Moreover, it is described in numerous works, e.g. [Hönig et al., 2016], [Ma et al., 2019], [Okumura et. al, 2022] how solutions of the ‘discertized idealized’ MAPF can be applied in practice and transferred to the real robots.

---

> ### Author Response · Authors · 2024-11-21
>
> Overall, we believe that POGEMA does not sweep under the carpet important real-world limitations (despite not considering them) but rather allows researchers and practitioners to focus their attention to the core task of any multi-agent navigation problem, i.e. to safe coordination of actions between the agents. In the same way, the other MARL navigation environments/benchmarks inherit the similar limitations, actually. For example, in the MAgent environment the actions are also discrete. In Flatland the agents are also confined to graphs, representing the railroad networks. In Nocturne the agents are confined to roads/lanes and the time horizon is very limited (9s). We believe that introducing such limitations is a reasonable trade-off between the real-world implications and the ability for the researchers to concentrate on the core problem of multi-agent navigation, i.e. coordination and cooperation between the agents.
>
> [Hönig et al., 2016] Hönig, W., Kumar, T.K., Cohen, L., Ma, H., Xu, H., Ayanian, N. and Koenig, S. Multi-agent path finding with kinematic constraints. ICAPS 2016. p. 477-485.
>
> [Ma et al., 2019] Ma, H., Hönig, W., Kumar, T.S., Ayanian, N. and Koenig, S. Lifelong path planning with kinematic constraints for multi-agent pickup and delivery. AAAI 2019. p. 7651-7658.
>
> [Okumura et. al, 2022] Okumura, K., Machida, M., Défago, X. and Tamura, Y. Priority inheritance with backtracking for iterative multi-agent path finding. Artificial Intelligence, 310, 2022. P.103752.
>
> W2: Thank you for your feedback. We have revised and rewritten the introduction to make it more concise.
>
> Q1: Thank you for pointing that out. We have corrected this typo in the revised version of the paper.
>
> Q2: Thank you for catching this error. It should indeed be “NP-Hard.”
>
> Q3: First, we agree that our definitions may lack rigor, but they capture the essence of the categorization. Your remarks are valid and insightful:
> - Regarding “the rewards of agents might not sum up to a fixed value but could have a relationship akin to being inversely proportional” — this aligns with our definition, as both describe the same principle: a joint strategy that benefits one player necessarily disadvantages others. We will refine our definition to clarify this point.
> - Regarding “in cooperative behaviors, agents do not necessarily share rewards, but they are working towards the same task” — working towards the same task implies a shared goal, which can be mathematically expressed as a “shared reward function.” We will include a remark to make this connection clearer.

---

> > ### Comment · Reviewer_iVSi · 2024-11-25
> >
> > Thank you for your response. I believe that most of my concerns have been addressed. As such, I will be increasing my score. Good luck!

---

> > > ### Author Response · Authors · 2024-11-25
> > >
> > > Thank you for your willingness to raise the score. We have updated the PDF file with the changes we promised to make in the general response:
> > > - We clarified the definitions of competitive and cooperative behaviors in lines 204-211, as per your suggestion.
> > > - We included steps-per-second performance metrics for POGEMA with a large population of agents in the Appendix (line 1615) to substantiate our claim of supporting >1000 agents. Specifically, we evaluate a maximum of 2048 agents using the Follower algorithm (as requested by Reviewer mDvH).
> > > - We incorporated a list of future research directions in the Conclusion section, as suggested by Reviewers mDvH and Jpm1.
> > > - We updated the abstract and title of the paper at the suggestion of Reviewer mDvH. Specifically, we removed the term “multi-agent navigation” from both the title and abstract and replaced it with the more specific term “multi-agent pathfinding.”
> > >
> > > However, based on your comment and the updated score, we wanted to ensure that we have fully addressed any remaining concerns that may have led to hesitation regarding the recommendation for acceptance. If there are still unresolved issues, we would appreciate the opportunity to address them before the discussion phase concludes.

---

### Official Review · Reviewer_mDvH · 2024-11-08

**Soundness:** 3
**Presentation:** 4
**Contribution:** 2
**Rating:** 6
**Confidence:** 3

**Summary:**

This paper introduces POGEMA, a benchmark for multi-agent pathfinding (MAPF) and its lifelong version (LMAPF) on grids. The contributions include a fast CPU-based environment, problem generators, visualization, and benchmarking tool for learnable, hybrid, and purely learned approaches.

**Strengths:**

The writing of this paper is crisp, and the visualizations are of excellent quality. The authors provide several examples and extensive code. While this is not the first paper to focus on MARL navigation, it is the first to fully focus on MAPF variants in a single repository. The proposed metrics are a much-needed tool to assess the performance of (L)MAPF approaches, not just based on the classical SoC/throughput but also on other metrics that can identify possible research directions.

**Weaknesses:**

1. I believe the title and abstract are misleading about the scope of the paper: while POGEMA appeals to a broad audience of MARL-based navigation in title and abstract, in fact it is about two variants of MAPF on discrete grids with simplified settings. For instance, in terms of MAPF, continuous variants like [1] does not appear to be considered. Moreover, there does not seem to be any mention about any-angle versions that would make the problem more realistic and interesting like [2].

2. One main concern about this paper is that while authors introduce a new MARL environment, they mostly perform inference with it and several results are not reproduced by retraining models. For instance, the DCC and SCRIMP codes do not contain any training script or guidance on how to reproduce results. I believe comparing training performance or speed with the proposed environment against the original implementation would strengthen the paper.

3. Authors claim scalability of >1000 agents. I cannot find any result to substantiate this claim.

4. The LaCAM version the authors mention refers to the first publication. However, new variants were released open source, i.e. LaCAM3 [3]. Moreover, versions as [4] can solve scales up to 10k agents in seconds.

5. Authors acknowledge the limitation of lack of JAX support and GPU-based parallelization tools. In this aspect, I believe the proposed (L)MAPF environments could be created by quite simple modifications of existing environments in Jumanji as the RobotWarehouse ([https://instadeepai.github.io/jumanji/environments/robot_warehouse/](https://instadeepai.github.io/jumanji/environments/robot_warehouse/)).

6. While this is a relatively minor point, I believe there are quite a few problems in terms of writing for the related work. Firstly: I don’t see why the paper should mention all multi-agent benchmarks, when the considered problems are only two very specialized ones. In this sense, Table 1 should contain information about “topic” or “area”, which is missing. GPU parallelization is not considered in the table. The main issue here is that it appears that POGEMA solves all the issues of previous MARL benchmarks such as Nocturne, while they solve arguably much simpler problems in restricted settings. Finally: I don’t see why there is a need to explain every single component as a separate paragraph for a total of almost 3 pages, such as “Python-based” and “PyPI listed" .

7. Finally, as a benchmark, it would be nice to include insights, i.e., what are possible future directions of research. Looking at the graph, one might conclude that there is no point in conducting research in MARL+MAPF, given that heuristic approaches perform well in all metrics, except scalability, only for the LMAPF case.


[1] Andreychuk, Anton, et al. "Multi-agent pathfinding with continuous time." Artificial Intelligence 305 (2022): 103662.

[1] Yakovlev, Konstantin, Anton Andreychuk, and Roni Stern. "Optimal and Bounded Suboptimal Any-Angle Multi-agent Pathfinding." Proceedings of the International Symposium on Combinatorial Search. Vol. 17. 2024.

[3] Okumura, Keisuke. "Engineering LaCAM\*: Towards Real-time, Large-scale, and Near-optimal Multi-agent Pathfinding." Proceedings of the 23rd International Conference on Autonomous Agents and Multiagent Systems. 2024.

[4] Okumura, Keisuke. "Improving lacam for scalable eventually optimal multi-agent pathfinding." arXiv preprint arXiv:2305.03632 (2023).

**Questions:**

1. One problem I have about the paper is more on the conceptual level, i.e. why the “PO” in POGEMA. “PO” stands for partially observable, and this makes sense in unknown environments that need exploration, such as the mentioned Nocturne benchmark. However, in the case of the proposed benchmark, I do not see why one should limit the observability as partial. Indeed, in my experience, the (L)MAPF problem arises in industrial settings in which maps are known a priori, which motivates the use of global heuristic controllers such as Lacam and RHCR. Why is partial observability used in such discrete settings, while SOTA heuristic approaches don’t use such a notion? Could leveraging full maps improve the performance of underperforming neural approaches?

2. What is the impact of sparse vs dense rewards in MARL for MAPF? Is it better to give +1 only at the end of the episode or a dense reward at each step?

3. In terms of conclusions for such a benchmark, it would be more interesting to identify the shortcomings of current models and help identify future research directions. What do you think these could be?

---

> ### Author Response · Authors · 2024-11-23
>
> W1: From the MAPF perspective POGEMA is indeed centered around what is called Classical MAPF [Stern et al., 2019] . This is a challenging problem which is known to be NP-Hard to solve optimally. Therefore, a large volume of research is focused on finding the best trade-off between the speed of obtaining a solution and the quality of this solution. Moreover, in one of the most important practical applications that drives the field, i.e. in automated warehouses, the assumption of ‘discretized grid with simplified setting’ holds. That justifies the value of concentrating on this setup. The core challenge of any multi-agent navigation problem, i.e. the need to coordinate the movements of the agents, is existent in the considered setup. Moreover, such an ‘idealized’ setup helps the researchers to better concentrate on the coordination/cooperation challenge. Still, we agree with the reviewer that some phrasings in the abstract/title/introduction may be altered.
>
> [Stern et al., 2019] Stern, R., Sturtevant, N., Felner, A., Koenig, S., Ma, H., Walker, T., Li, J., Atzmon, D., Cohen, L., Kumar, T.K. and Barták, R.. Multi-agent pathfinding: Definitions, variants, and benchmarks. In SoCS 2019.
>
> W2: We appreciate the reviewer’s feedback and agree that providing training code for DCC and SCRIMP would enhance reproducibility. In our work, we used pre-trained weights provided by the authors of these approaches, as their training relied on custom frameworks and environments not compatible with POGEMA. For instance, SCRIMP approach has a collision-resolution technique integrated with its environment, which is not supported by POGEMA, and we would like to avoid implementing such ad-hoc techniques on the level of the environment that violate decentralized decision-making. While we acknowledge the value of comparing training performance in a unified setting, we opted for a different evaluation approach: allowing training on diverse scenarios and evaluating on a hold-out dataset.
>
>
> W3: We thank the reviewer for highlighting this concern. While we did not include experiments scaling to >1000 agents in the initial submission—primarily because many communication-based approaches cannot handle such numbers within feasible time—it is straightforward to test scalability with faster algorithms, such as Follower.
>
> To address this issue, we will expand the POGEMA Speed Performance Evaluation section to include steps-per-second comparisons for setups with more than 1000 agents. This will illustrate how processing time changes as the number of agents increases. We will update the manuscript accordingly during the discussion period.
>
> As an intermediate response, we have provided an animation [here](https://anonymous.4open.science/r/pogema-7439/pogema-ep00001-large-validation-mazes-seed-0-seed0.svg) showing Follower running 1024 agents in a single environment of size 128×128.
>
> W4: In our code, we used the latest version at the time or writing, i.e. LaCAM-v3. Thank you for bringing this to our attention; we have updated the references to include this citation as well. Additionally, we have revised the appendix, extending the Evaluation Setup Details section to include links to the GitHub repositories of the code used.
>
> W5: Indeed, lacking JAX support and GPU-based parallelization tools is a limitation. However, while many new environments are specifically designed for GPU acceleration, we argue that this may not always be necessary. Once an environment can process a sufficient number of steps per second across parallel simulations, the bottleneck often shifts from simulation to neural network computation. By avoiding reliance on GPUs or TPUs for simulation, POGEMA keeps these resources fully available for training. Additionally, we extended the Speed Performance Evaluation section in the Appendix, where we compared POGEMA to JaXMARL. Notably, POGEMA outperforms Gigastep (another environment, suggested by reviewer iVSi)  in reported throughput, achieving a maximum of 3.1 million observations per second on CPU.
>
> Regarding the Robot Warehouse environment (referred to as RWARE in Table 1) suggested by the reviewer, it is already included in our comparison. After analyzing its implementation in JAX, we identified challenges in adapting it to a rich (L)MAPF setting. The primary issue lies in procedural generation. The current implementation assumes a warehouse pattern with a single connected component, allowing targets to be placed at any available slot. However, in (L)MAPF, there may be multiple independent components, and it is critical to ensure that each target and agent belong to the same connected component. This typically requires BFS-based preprocessing, which is difficult to parallelize efficiently on GPUs. Furthermore, almost all learnable MAPF approaches use target guidance, which is often constructed using the A* algorithm or precomputed cost-to-go values. These computations are also difficult to parallelize efficiently on GPUs.

---

> ### Author Response · Authors · 2024-11-23
>
> W6: Regarding the inclusion of all multi-agent benchmarks, the intention behind Table 1 is not to claim that the listed issues are the only challenges in MARL or multi-agent systems. Instead, the table highlights key problems addressed by the presented approach and contextualizes POGEMA within the broader landscape of MARL benchmarks. Many of these issues are indeed important and require further exploration.
>
> On the topic of GPU parallelization, we acknowledge that it was not explicitly addressed in the table. However, we have included a proxy column reflecting how quickly environments can generate samples, which provides an indirect measure of computational efficiency. While GPU parallelization is an important factor, CPU parallelization is equally relevant for many applications, and our focus on this aspect reflects the flexibility of POGEMA. We will clarify this in the manuscript to ensure a balanced representation of parallelization methods.
>
> Finally, regarding the detailed explanations in the related work section, we aimed to provide a comprehensive discussion of the components and context for POGEMA. However, we agree that some parts could be condensed for clarity and brevity. We will revise the related work section to reduce redundancy and streamline the descriptions, ensuring the focus remains on the most relevant comparisons and contributions.
>
> W7-Q3: Thank you for pointing this out. Here are several future research directions that we would like to emphasize and include in the paper:
> 1. Large scale MARL:
> Many MARL approaches underperform compared to learnable MAPF methods, even when using preprocessing techniques from second (e.g., from Follower). This suggests that POGEMA presents a compelling challenge for large-scale MARL training setups, particularly for the CTDE (Centralized Training, Decentralized Execution) paradigm. Popular methods like QMIX and MAMBA struggle to scale to large numbers of agents in such setups. In our experiments, we were unable to train these methods on the same maps and with the same number of agents as in Follower (which uses independent PPO), where the authors trained agents in environments with up to 256 agents.
> 2. Large-scale imitation learning:
> POGEMA provides a fast environment and efficient centralized baselines that can be used to generate high-quality data for training. This is particularly useful for foundation models, as the procedural generation of maps allows for an unlimited supply of expert data.
> 3. Communication learning:
> POGEMA’s large maps inherently require agents to rely on local communication, making it an ideal testbed for MARL approaches focusing on communication. While communication has been extensively studied in MARL, its application in large-scale settings remains underexplored, and POGEMA provides a platform for advancing this field.
> 4. Memory-efficient methods:
> Most existing approaches (except for Switcher) rely on global guidance to reach goals in POGEMA, which poses significant challenges in multi-agent scenarios requiring memory. The sharp drop in performance compared to  methods, which uses target guidance, highlights the need for memory-efficient strategies, making this an important area for further exploration.
> 5. Heterogeneous policies (opponent / ally modelling)
> Currently, no centralized approaches can effectively handle scenarios with multiple policies. Learnable methods have the potential to close this gap, enabling agents to coordinate effectively in heterogeneous settings. This also opens an exciting avenue for studying how different algorithms can be trained concurrently in the same environment, fostering advancements in collaborative and adversarial multi-agent interactions.

---

> ### Author Response · Authors · 2024-11-23
>
> Q1: First, partial observability is conceptually applicable to any learnable MAPF approach because an agent’s policy typically relies only on its local observation. While it is possible to provide the policy with the full state in some simple setups (due to limited size of network input), doing so is unnecessary and diminishes many of the advantages of the partially observable case. Partial observability promotes scalability, decentralization, and robustness, which are critical in dynamic and multi-agent environments.
>
> Second, while it is true that maps are often known a priori in industrial settings like warehouses, this assumption limits adaptability. Traditional heuristic approaches used in these domains are tailored for static environments and may struggle with dynamic changes, such as temporary obstacles or blocked paths. In contrast, learnable decentralized algorithms can adapt to such changes. Exploring these types of approaches could broaden their applicability to more dynamic and uncertain settings beyond traditional use cases.
>
> Third, regarding map availability, most learnable MAPF approaches assume the map is already known (e.g. Follower, DCC, SCRIMP etc).. However, there are algorithms, such as the “Switcher” approaches in the benchmark, that can construct or update the map during inference. If the map is assumed to be known beforehand, many approaches use guidance, such as providing the algorithm with partial information about local shortest paths to the goal, which significantly improves their performance. Even with access to this information, these approaches lack of global knowledge of other agents’ positions or their target locations. This limitation aligns with the decentralized, partially observable paradigm, fostering robustness and generalization to scenarios where global information is unavailable or unreliable.
>
> Thank you for the question. Dense rewards can make the learning process easier for RL algorithms, especially when they face exploration challenges in environments with long planning horizons. For POGEMA, we chose to use a sparse reward of +1 at the end of the episode as the default. However, for benchmarking and training MARL algorithms, we used reward structures inspired by Follower to facilitate learning and improve performance.
>
> Your question, while seemingly simple, touches on a deeper issue. Dense rewards can introduce bias, as agents may exploit the reward signal or solve unintended tasks without optimizing the main objective. This can be seen in many RL approaches for learnable MAPF, where reward shaping schemes are often introduced to guide the agent’s behavior.
>
> When we move beyond centralized training to consider decentralized settings, the reward function becomes even more critical. In multi-agent scenarios, tasks can become adversarial—optimizing one agent’s objective may reduce rewards for others. Designing reward functions in these cases is particularly challenging. For centralized training, the issue shifts to scalability. As the agent population grows, so do the state spaces, and with many agents influencing the global reward, this greatly complicates training.

---

> > ### Comment · Reviewer_mDvH · 2024-11-25
> >
> > Thanks for your reply.
> >
> > I will add follow-up remarks and questions:
> >
> > > Still, we agree with the reviewer that some phrasings in the abstract/title/introduction may be altered.
> >
> > Do you plan to alter them, and if so, how? I cannot see any modification in the modified paper. I still believe the paper appeals to a broad audience of multi-agent navigation, but in practice, it is about arguably overly simplified settings.
> >
> > > By avoiding reliance on GPUs or TPUs for simulation, POGEMA keeps these resources fully available for training.
> >
> > To my knowledge, the biggest limitation of RL is the data transfer between CPU and GPU. In this sense, I do not see how liberating GPU cores (which are plenty) would help give this bottleneck since models in POGEMA are on GPU and environments on CPU. Do you have some insight on this?
> >
> > > W7-Q3: Thank you for pointing this out. Here are several future research directions that we would like to emphasize and include in the paper
> >
> > Thanks for providing them. I believe such directions would also be beneficial to have in the paper itself.
> >
> > > Q1: First, partial observability is conceptually applicable to any learnable MAPF approach because an agent’s policy typically relies only on its local observation. While it is possible to provide the policy with the full state in some simple setups (due to limited size of network input), doing so is unnecessary and diminishes many of the advantages of the partially observable case. Partial observability promotes scalability, decentralization, and robustness, which are critical in dynamic and multi-agent environments.
> >
> > While I agree with the importance of decentralization, I still do not see how this is studied in the benchmark and why this should be assumed. In particular, SOTA heuristics, such as LACAM and RHCR, are centralized, and those are the main baselines for all approaches. This indicates to me, as a reader, that centralized approaches should be used in real-world settings. I would appreciate it if you could point out references/case studies of MAPF in industrial settings which assumes partial observability.

---

> > > ### Author Response · Authors · 2024-11-25
> > >
> > > We thank the reviewer for their involvement in the discussion. Below are our responses to the follow-up questions and remarks.
> > >
> > > > Do you plan to alter them, and if so, how? I cannot see any modification in the modified paper. I still believe the paper appeals to a broad audience of multi-agent navigation, but in practice, it is about arguably overly simplified settings.
> > >
> > > We have now modified the title and abstract of the paper following the discussion. Specifically, we have removed the term “multi-agent navigation” from both the title and the abstract and replaced it with the more specific term “multi-agent pathfinding.”
> > >
> > > > To my knowledge, the biggest limitation of RL is the data transfer between CPU and GPU. In this sense, I do not see how liberating GPU cores (which are plenty) would help give this bottleneck since models in POGEMA are on GPU and environments on CPU. Do you have some insight on this?
> > >
> > > Yes, CPU-GPU data transfer bottlenecks are a common challenge in RL and MARL. However, modern frameworks like SampleFactory and PufferLib address this by employing fixed memory allocation and asynchronous sampling and training. For instance, SampleFactory handles data transfer within rollout workers, ensuring that experience is preemptively transferred to GPU memory and ready for training. This approach minimizes delays during batch sampling. For online methods like PPO, discrepancies between the data-generation policy and the current training policy can cause lags. V-Trace addresses this by correcting value estimates, ensuring stable and efficient training in asynchronous settings.
> > >
> > > In the context of POGEMA, using CPUs for environment simulation frees GPUs entirely for training. This separation allows GPUs to focus solely on forward and backward passes, avoiding the context-switching overhead that occurs when GPUs handle both simulation and training. Additionally, this approach reserves more GPU memory for larger models or higher batch sizes by offloading memory-intensive simulation tasks to CPUs. While data transfer between CPU and GPU remains a factor, frameworks leveraging asynchronous operations can mitigate this bottleneck, ensuring that CPU-based simulation does not hinder GPU training efficiency.
> > >
> > > Notably, JIT compilation, available through tools like jax.jit or torch.compile, imposes restrictions on the design of the environment, making branching more difficult and often suggesting adherence to the functional programming paradigm. Furthermore, it complicates tasks such as procedural generation or environment modification for curriculum learning or lifelong learning, due to the necessity of compiling the entire environment.
> > >
> > > > Thanks for providing them. I believe such directions would also be beneficial to have in the paper itself.
> > >
> > > Thank you for your feedback! We have incorporated the suggested future research directions into the conclusion section of the paper.
> > >
> > > > While I agree with the importance of decentralization, I still do not see how this is studied in the benchmark and why this should be assumed. In particular, SOTA heuristics, such as LACAM and RHCR, are centralized, and those are the main baselines for all approaches. This indicates to me, as a reader, that centralized approaches should be used in real-world settings. I would appreciate it if you could point out references/case studies of MAPF in industrial settings which assumes partial observability.
> > >
> > > While centralized approaches are still used in industrial settings (according to our internal communications, which we cannot reference here), their costs increase significantly as the number of agents grows. Decentralized systems present a promising alternative, often implying partial observability and communication, which has gained considerable attention in both academia and industry. Many papers, starting with the seminal PRIMAL work, have been emerging from research groups in contact with industrial players, highlighting the relevance of decentralized approaches.
> > >
> > > A relevant example involving RHCR is provided in the Learn to Follow paper (Skrynnik et al., 2024a), where the authors compare the performance of Follower and RHCR in a warehouse environment under two different setups with limited decision-making time. In the 10-second time limit scenario, RHCR outperformed Follower. However, with a 1-second time limit, RHCR struggled to find a plan, underperforming significantly compared to Follower. This experiment highlights the limitations of centralized approaches like RHCR in scenarios with tight decision-making constraints, which are common in real-time systems.

---

> > > > ### Comment · Reviewer_mDvH · 2024-11-26
> > > >
> > > > I thank the authors for their further clarifications.
> > > >
> > > > My major issue was that the paper appealed to a broader audience than its actual content.
> > > >
> > > > However, if POGEMA adjusts the scope and title, as the authors have changed the title to "POGEMA: A BENCHMARK PLATFORM FOR COOPERATIVE MULTI-AGENT **PATHFINDING**," I do not have other major issues with it, and the benchmark is indeed technically solid.
> > > >
> > > > Thus, I will raise my score.

---

### Author Response · Authors · 2024-11-23
**General Response**

We sincerely thank the reviewers for their thoughtful feedback, which will help refine our work. We appreciate the recognition of POGEMA’s computational efficiency, scalability, and ability to generate diverse, procedurally-created problems, making it valuable for evaluating both MARL and MAPF algorithms (Reviewers Jpm1, iVSi, and mDvH). The acknowledgment of its fast performance and capacity for large-scale agent training and evaluation highlights its practicality (Reviewers iVSi and Jpm1). We are grateful for the positive remarks on the clarity of writing, high-quality visualizations, and extensive code examples (Reviewer mDvH), as well as the thorough experimental studies and metrics proposed to guide future research (Reviewers Jpm1 and mDvH).

We’ve answered specific reviewers question in responses to the. Here we provide a list of changes in manuscript:
- The Introduction section has been updated, as suggested by Reviewer iVSi, to make it more concise and better position the paper.
- We have added Magent and Gigastep to Table 1 and will include a description comparing these benchmarks to POGEMA in the corresponding section of the Appendix, as outlined in our response to Reviewer iVSi.
- The Speed Performance Evaluation section in the Appendix has been extended to compare POGEMA with JaXMARL, and additional discussion on GPU acceleration has been included.


During the discussion period, we commit to incorporating the following updates:
- Clarify the definitions of competitive and cooperative behaviors in lines 221–225, as suggested by Reviewer iVSi.
- Include steps-per-second performance metrics for POGEMA with a large population of agents in the Appendix to substantiate our claim of supporting >1000 agents (Reviewer mDvH).
- IIncorporate a list of future research directions in the Conclusion section, as suggested by Reviewers mDvH and Jpm1. These directions are detailed in our responses to the corresponding questions.

---

> ### Author Response · Authors · 2024-11-28
> **General Response (follow-up)**
>
> Following discussions with Reviewers Jpm1 and iVSi, we have significantly revised the introduction to make it more focused and streamlined (please see the revised pdf). Additionally, we have made slight adjustments to the wording of the title and abstract. We agree with the reviewers’ observation that our work focuses on a specific variant of the multi-agent navigation problem, namely the one that employs discretized atomic actions and known as multi-agent pathfinding in the literature. However, we emphasize that the core challenge of any multi-agent navigation problem, the need to coordinate actions to avoid conflicts, remains central to the formulation we study. We are pleased to note that the reviewers do not appear to dispute this argument.
>
> Finally, we confirm that all updates we committed to during the discussion period have been implemented.

---

### Meta-Review · Area_Chair_YJ6B · 2024-12-21

**Metareview:**

The paper introduces POGEMA, a novel benchmark platform for cooperative multi-agent pathfinding (MAPF), which provides a fast, scalable environment, procedural generation of problems, and well-defined evaluation protocols. It effectively bridges the gap between classical heuristic methods, learning-based approaches, and hybrid techniques, enabling fair and reproducible evaluations. The benchmark's ability to handle up to 1 million agents and its comprehensive experimental comparisons highlight its strong alignment with the datasets and benchmarks track criteria. While some reviewers questioned its novelty, POGEMA's technical robustness and clear utility for advancing large-scale MARL and MAPF research validate its acceptance.

**Additional Comments On Reviewer Discussion:**

Reviewers raised concerns about the paper's scope, computational efficiency, and real-world relevance.

Reviewer *iVSi* highlighted that the benchmark's contributions were insufficiently distinguished from existing platforms like Gigastep. Reviewer *mDvH* critiqued the overgeneralization in the title and abstract, prompting the authors to revise these and emphasize POGEMA's focus on MAPF.

Concerns about scalability claims were addressed with experimental results showing support for up to 1 million agents, and the authors incorporated future research directions to clarify POGEMA’s broader impact. Despite minor remaining reservations from some reviewers, these revisions and clarifications satisfied the majority, leading to an overall recommendation for acceptance.

---

### Decision · Program_Chairs · 2025-01-22

Accept (Poster)